# Optimization of the Borehole Wall Protection Slurry Ratio and Film-Forming Mechanism in Water-Rich Sandy Strata

Xiaodong Liu [1], Meng Li [1,2], Peiyue Qiu [3], Liyun Tang [4,*], Zhenghong Liu [5] and Fusheng Zhang [4]

1 China Railway First Survey and Design Institute Group Ltd., Xi'an 710043, China
2 Shanxi Railway and Underground Traffic Engineering Key Laboratory (FSDI), Xi'an 710043, China
3 Zhengzhou Urban and Rural Construction Bureau, Zhengzhou 450006, China
4 School of Architecture and Civil Engineering, Xi'an University of Science and Technology, Xi'an 710054, China
5 China Jikan Research Institute of Engineering Investigations and Design Co., Ltd., Xi'an 710043, China
* Correspondence: tangly@xust.edu.cn; Tel.: +86-029-8558-3153

**Abstract**

Conventional slurry wall protection exhibits reduced film performance upon exposure to water in saturated sand layers with high permeability, frequently resulting in hole wall instability. Optimizing the slurry ratio to enhance film performance is thus critical for borehole stability. A multiple regression model was developed to determine the optimal slurry ratio for saturated sand. Slurry permeability tests assessed filtration loss, film formation time, and film morphology changes. Scanning electron microscopy (SEM) further elucidated the film formation mechanism. Bentonite, clay, $Na_2CO_3$, and sodium carboxymethyl cellulose (CMC) significantly affected the slurry's properties: specific gravity and sand content increased with bentonite/clay; viscosity increased with CMC; and pH increased with $Na_2CO_3$. The optimized slurry (water–bentonite–$Na_2CO_3$–clay–CMC = 1000:220:32:110:1; specific gravity, 1.20 $g/cm^3$; viscosity, 29 s) demonstrated low filtration loss and stable film morphology. SEM revealed that simultaneous CMC and clay addition (ratio of 1:110) improved film surface flatness, reduced porosity and pore size, enhanced formation surface filling, and produced a denser film. The optimized slurry ratio significantly enhanced film performance in saturated sand layers. The findings provide a theoretical and engineering framework for bored pile wall protection slurry design and film formation mechanisms.

**Keywords:** slurry wall protection; multiple regression model; response surface analysis; film-forming mechanism; scanning electron microscope

## 1. Introduction

Globally, bored pile technology is widely used for bridge foundation construction. However, in saturated sand layers with weak cementation, this method often leads to borehole instability issues, such as wall shrinkage and collapse [1], significantly impacting construction schedules and long-term structural performance. Slurry wall protection serves as the primary technique for maintaining borehole stability during drilling operations. However, in saturated sand formations, multiple factors, including soil characteristics, pore water pressure, borehole depth, and slurry film morphology, collectively contribute to a complex mechanism. Conventional bentonite-based slurries are particularly susceptible to the adverse effects of saturated sands, resulting in compromised wall protection performance [2]. Simultaneously, the influence mechanism of the particle infiltration process on the formation of slurry film under different slurry ratios is not clear, which makes it difficult to guarantee the hole quality and long-term stability of bored piles. Therefore,

based on the special requirements of saturated sand strata for slurry characteristics, the influence law of different additives is analyzed by a multiple regression model to optimize the current wall-protection slurry formulation. The microstructure change characteristics of the slurry film are clarified by SEM, and the internal mechanism of slurry permeation film formation in saturated sand strata is further analyzed by combining the test results. The results can provide theoretical and engineering guidance for the slurry proportion scheme and the penetration film formation mechanism in saturated sand strata.

Recent studies reveal that drilling fluid optimization has evolved toward three objectives: high water retention, low filtration loss, and rapid filter cake formation. Li et al. [3] employed discrete element modeling (DEM) to investigate microcrack effects on single-particle compressive strength in coarse aggregates, demonstrating that microcrack quantity, location, and orientation significantly influence the Weibull distribution and compressive strength. Gupta et al. [4] explored kaolin stabilization via geopolymerization, achieving substantial improvements in unconfined compressive strength and dynamic properties of soft soils through optimized precursor/alkaline activator ratios. Zhou et al. [5] developed a modified ultrafine cement-based grout (MUCG), determining its optimal formulation via orthogonal experiments to enhance fluidity and early strength. Zhang et al. [6] systematically evaluated bentonite–polymer composite rheological enhancements, noting significant filter cake brittleness at excessive polymer dosages. Ding et al. [7] established a rheology–formation coupling model for seawater-resistant slurries, though its applicability to water-rich sand strata remains unverified due to testing on cohesive soils. The key parameters characterizing the characteristics of a slurry solution are slurry specific gravity, viscosity, and filtration loss [8–12], and the slurry penetration film test directly assesses the adaptability of the slurry to the formation. The factors affecting the slurry penetration film include formation conditions [13–15] and slurry additives [16,17]. Rita et al. [18,19] investigated the effects of $Na_2CO_3$, sodium carboxymethyl cellulose (CMC), and clay on the specific gravity and viscosity of a slurry. Their findings indicate that the specific gravity and viscosity of the slurry increased with the increase in the clay and CMC content, and CMC and $Na_2CO_3$ had a thickening effect on the slurry.

In the realm of nanomaterial applications, Edalatfar et al. [19] demonstrated that nano-additives significantly reduce slurry filtration loss by pore-filling effects, though their long-term stability under field conditions remains underexplored. Similarly, Yang et al. [20] achieved a 1.4-fold increase in the bentonite swelling index using sodium hexametaphosphate, highlighting the potential of chemical modifiers—yet their cost-effectiveness for large-scale projects warrants further investigation. The shift toward sustainable additives has yielded promising alternatives. Al-Hameedi et al. [21] investigated the impact of incorporating 0.5% to 3.5% black sunflower seed shell powder into drilling mud, demonstrating that this additive, across various particle sizes, effectively modifies viscosity, controls filtration loss, and enhances wellbore stabilization. Bagum et al. [22] identified Aloe Vera as a non-toxic substitute for conventional chemicals. Lei et al. [2] further expanded this trend by proving agar/guar gum's superior film-forming ability in sandy strata. However, these studies predominantly focus on laboratory-scale performance, leaving scalability and field compatibility gaps unresolved. For performance optimization, AlAwad et al. [23] standardized cement–rock bonding tests, emphasizing the need for region-specific slurry designs, whereas Nashed et al. [24] pioneered machine learning for bottom-hole pressure prediction—a methodological leap that nonetheless requires validation across diverse geological settings. Chen et al. [25] conducted scanning electron microscopy (SEM) analysis, revealing that hydrophobically thermo-thickening polymer molecules undergo complete extension within the slurry matrix and form effective bonds with hydration products, thereby promoting uniform microstructure development in the hardened slurry. Through

SEM images, Qin et al. [26] observed that a slurry film exposed to seawater intrusion contained many well-connected agglomeration channels, which reduced its sealing quality. Gao et al. [27] conducted SEM analysis of a slurry film, revealing that the additives improved film compactness, enhanced the protective properties of the modified slurry, and reduced filtration loss. Collectively, there remain three critical gaps: (1) the absence of quantitative mix-design models specifically for water-rich, highly permeable, cohesionless sand strata; (2) the insufficient elucidation of micro-scale pore evolution mechanisms in filter cakes under CMC–clay synergy; and (3) the lack of integrated design methodologies correlating macro-performance, microstructure, and field conditions.

This study takes saturated sand layers as the engineering context, examining the impact of various polymer additives on slurry properties while establishing clay as the fundamental solid-phase material. The optimal slurry ratio was determined through multiple regression modeling, incorporating key parameters including specific gravity, viscosity, slurry film thickness, and filtration loss. Permeability tests were conducted to validate the optimized formulation. The slurry permeation and film-forming mechanism were elucidated based on the effective stress principle and rheological pore-blocking effects. The findings provide both a theoretical foundation and practical guidance for borehole wall protection in bored pile construction within saturated sand strata. The key innovation of this study involves not simply using conventional clay components but systematically optimizing their synergistic interaction with CMC. Rigorous ratio control (clay–CMC = 110:1) combined with SEM-based mechanistic analysis reveals how this specific combination significantly enhances film density and pore-filling efficiency, thereby addressing persistent challenges in saturated sand stratum applications. This study aims to (1) establish a quantitative model linking additives to slurry properties using RSM; (2) reveal the microscopic mechanism of film enhancement via SEM; and (3) provide field-applicable ratios for saturated sand layers, addressing the gap in existing codes.

## 2. Optimization Scheme of the Slurry for Wall Protection of Saturated Sand Layers

### 2.1. Project Overview and Slurry Performance

#### 2.1.1. Project Overview

This project is situated in the lower Weihe River basin, encompassing the first terrace of the Weihe River and the floodplain area of the Weihe–Jinghe River confluence. The site features thick sand layers with weak cementation, high permeability, and poor formation stability. The combination of negligible cohesion and elevated pore water pressure in the sand stratum renders the generalized parameter ranges stipulated by Chinese pile construction codes (DZ/T0155-95 [28] and JGJ/T 225-2010 [29]) inadequate for the site-specific conditions encountered in this project's saturated alluvial sands. This poses potentially significant safety hazards to the pile foundation hole-forming construction of the project, demanding enhanced performance of the slurry characteristics in the hole-forming construction. In order to ensure the safety of drilling construction, the slurry preparation test was conducted. Considering the resource utilization and cost savings of stratum clay, clay and polymer materials were selected as slurry additives. The polymer materials consisted of polyacrylamide (PAM) and CMC, which were new additives to the slurry. A multiple linear regression analysis was conducted to evaluate the effects of various additives on slurry performance. Permeation tests were performed based on experimentally measured slurry properties. By systematically comparing the film formation time, filtration loss, and membrane morphology across different preparation protocols, optimal thresholds for key slurry parameters were established. This methodology provided an empirically validated slurry formulation for saturated sand formations in engineering practice.

2.1.2. Basic Properties of the Slurry

In drilling construction, a slurry serves four critical functions: (1) stabilizing the borehole wall by forming a low-permeability slurry film, (2) balancing formation pressure to prevent collapse, (3) lubricating drilling tools to reduce wear, and (4) removing cuttings to maintain hole cleanliness. These functions are optimized through tailored slurry ratios, particularly in saturated sand strata where conventional slurries underperform due to high permeability and weak cementation.

## 2.2. Performance Index and Influencing Factors of the Wall Protection Slurry

2.2.1. Slurry Performance Index

In the process of bored pile formation, the stability of the hole wall relies on maintaining the specific gravity and viscosity of the slurry in the hole. As the main method to ensure the stability of the hole wall of the bored pile, the performance index of the slurry solution directly impacts the quality of the hole. Among the indicators, the specific gravity, viscosity, slurry film thickness, and filtration loss of the slurry demonstrate significant influence on the quality of the hole. The specific indicators are shown in Table 1.

**Table 1.** Main performance indexes of slurry *.

| Specific Gravity (g/cm$^3$) | Viscosity (s) | Sand Content (%) | pH |
| --- | --- | --- | --- |
| 1.20–1.40 | 20–45 | <4 | 8–12 |

* The recommended parameters for drilling slurry are as follows: specific gravity: 1.14–1.25 g/cm$^3$ [28]; viscosity: 20–45 s [28,30]; and sand content control threshold: <4% [31].

Specific gravity: Specific gravity controls solid particle concentration to balance formation pressure. Low values risk stratification and poor film formation, while high values increase tool wear and reduce drilling efficiency. Specific gravity was measured using a lever-action mud balance (Model: XYZ-100; OFI Testing Equipment, Inc., Houston, TX, USA) following API RP 13B-1 [32]. The procedure included horizontal calibration, steady filling to the datum line, and sealing to eliminate surface tension. Readings were taken at torque equilibrium, with triplicate measurements (mean deviation < 1%).

Viscosity: Viscosity determines shear resistance and film-forming capacity, which are essential for mitigating hole collapse and particle settlement in permeable strata. The viscosity of the slurry in this test was determined using the Marsh funnel viscometer (Model: 1500-10; OFI Testing Equipment, Inc., Houston, TX, USA), which measures the time (in seconds, s) required for 500 mL of the slurry to flow through the funnel. This is a widely adopted empirical method in drilling engineering for quality control. The apparent viscosity of the slurry was determined to be 15 mPa·s using a Fann 35 rheometer (Model: Fann 35; Fann Instrument Company, Houston, TX, USA) at 300 rpm, which correlates with a Marsh funnel viscometer time of 29 s for 500 mL of slurry.

Sand content: Excessive sand (>4%) compromises lubrication and equipment longevity, whereas insufficient sand content reduces cuttings transport. The sand content was quantified via sieve analysis (model: NA-1 sand content set) per ASTM D4380, involving iterative washing/filtration (0.074 mm sieve) and sedimentation to isolate particles > 74 μm.

pH: An alkaline environment optimizes bentonite dispersion and slurry stability while inhibiting bacterial growth. pH was determined colorimetrically using MColorpHast strips (Manufacturer: Shanghai Yuanhu Industrial Co., Ltd., Shanghai, China; ISO 3071-compliant [33]) with a resolution of ±0.5 pH. The strips were immersed in homogenized slurry, and the stabilized color was matched to a reference card. pH was recorded as integers due to the ±0.5 pH resolution of the colorimetric strips.

2.2.2. Influence Factors of the Slurry Performance Index

The slurry is primarily composed of water, clay, and bentonite. According to the characteristics of different sites, pore-forming methods, and equipment configuration, the influence of different ratio components on the performance index of the slurry is significant. The influence of bentonite, clay, CMC, and $Na_2CO_3$ on the performance index of the slurry is as follows:

(1) Bentonite (sodium type): Sourced from Zhejiang Hongyu New Materials Co., Ltd. (Huzhou, China), bentonite serves as the primary solid-phase material in the slurry system. Its high hygroscopicity and adsorption capacity enable it to absorb water and various inorganic substances, forming colloidal particles that seal gaps and fractures to prevent fluid loss. Additionally, bentonite increases slurry density and viscosity, thereby aiding in pore pressure control and borehole wall stabilization. As an economical and environmentally benign material, it offers both technical and operational advantages.

(2) Clay: Collected from the construction site of the Weihe River, the clay particles exhibit adsorption and hydration properties, which enhance the stability of the slurry dispersion system. By adjusting the clay content, varying adsorption and hydration effects can be achieved, thereby producing slurries with distinct properties.

(3) $Na_2CO_3$: Certified as food-grade by Binhu, $Na_2CO_3$ primarily functions to neutralize organic acids and acidic gases, thereby mitigating slurry corrosion. Additionally, it maintains slurry fluidity and prevents water loss.

(4) CMC: Produced by Chongqing Lihong Fine Chemicals Co., Ltd. (Chongqing, China), this polymeric organic material functions as a viscosifier to mitigate excessive slurry water loss. Additionally, it exhibits colloidal protective properties and serves as an engineering material to prevent slurry contamination.

(5) PAM: Produced by Chongqing Lihong Fine Chemicals Co., Ltd., these water-soluble polymers exhibit dissolution characteristics influenced by molecular weight, ionic type, and particle fineness. Complete dissolution requires prolonged stirring. The aqueous solutions demonstrate high viscosity, with viscosity positively correlated to molecular weight. While stable at room temperature, the polymers undergo thermal degradation at elevated temperatures, resulting in viscosity reduction.

*2.3. Optimization Analysis of Slurry Polymer Additives*

To evaluate the influence of different polymer additives on slurry properties, the slurry polymer additive test was first carried out. The field slurry ratio of water–bentonite–$Na_2CO_3$ = 1000:220:32 was used as a comparison, and 100 g of clay was used as an additive to improve slurry characteristics. The slurry solution was mixed with the slurry solution. The slurry characteristics were mainly based on specific gravity, viscosity, sand content, and pH, and the effects of the formulations on the slurry characteristics were compared and analyzed. The effect of the polymers (CMC, PAM) on the slurry characteristics is shown in Table 2.

Table 2 presents the variations in slurry performance parameters under different polymer treatments. Regarding slurry specific gravity, both CMC and PAM exhibit similar trends with increasing dosages, showing a comparable increase from 1.14 $g/cm^3$ to 1.20–1.21 $g/cm^3$, suggesting that their influence on slurry density is nearly equivalent. In contrast, viscosity measurements reveal a significant divergence between PAM and CMC as dosage increases. At a 1 g dosage, the viscosity difference is minimal (3 s), but at 4 g, PAM demonstrates substantially higher viscosity (64 s) compared to CMC. This highlights PAM's superior thickening efficiency at equivalent mass concentrations. Under identical additive concentrations, the sand content of the PAM-modified slurries was significantly

higher compared to the CMC-modified slurries, with the absolute difference consistently maintained at approximately 1 percentage point. This phenomenon originates from the coarsening of solid particles induced by PAM's flocculation effect, as evidenced by the measured data in Table 2. During slurry preparation, visual and microscopic observations demonstrate that polyacrylamide (PAM) effectively flocculates solid particles, enhancing their dispersion in the aqueous phase while significantly reducing water loss. However, this flocculation simultaneously increases slurry viscosity and promotes particle agglomeration, resulting in coarser microstructures and a measurable increase in sand content. Notably, at the 4 g dosage threshold, the PAM-generated flocculates exhibited particle retention on 200-mesh sieves ($\geq$75 µm), indicating compromised injectability. Comparative analysis of rheological and filtration characteristics consequently identifies carboxymethyl cellulose (CMC) as the superior polymeric additive for field applications, as it maintains an optimal balance between viscosity control and filtration stability without inducing excessive particle growth. This conclusion aligns with practical engineering requirements for pumpable, stable slurries in saturated sand formations.

**Table 2.** Slurry parameters under the action of different polymers.

| Number | CMC (g) | PAM (g) | Specific Gravity (g/cm$^3$) | Viscosity (s) |
|---|---|---|---|---|
| 1 | 0 | | 1.14 | 19 |
| 2 | 1 | | 1.20 | 25 |
| 3 | 2 | 0 | 1.20 | 29 |
| 4 | 3 | | 1.20 | 44 |
| 5 | 4 | | 1.20 | 49 |
| 6 | | 1 | 1.20 | 28 |
| 7 | 0 | 2 | 1.21 | 44 |
| 8 | | 3 | 1.21 | 76 |
| 9 | | 4 | 1.21 | 113 |

*2.4. Optimization Analysis of Slurry Ratio Parameters*

The Box–Behnken design is an efficient response surface methodology that requires fewer experimental runs than full factorial designs while maintaining the ability to estimate quadratic effects [34]. This design is particularly suitable for optimizing multiple slurry parameters. In this investigation, the F-value was primarily employed in the analysis of variance (ANOVA) for multivariate regression models. By comparing the mean square of factors with the mean square of error, it quantifies the significance of individual factors (e.g., silty clay, CMC) and interaction terms on response variables (e.g., drilling fluid specific gravity, viscosity). A higher *F*-value indicates greater statistical significance of a factor's impact on the response variable, thereby indirectly reflecting its sensitivity. This approach aligns with classical methodologies in experimental design (as documented in Reference [35]), wherein significance testing determines variable importance [36,37].

In order to study the changes in the slurry performance indexes, such as specific gravity, viscosity, sand content, and pH, under the multi-factor combination ratio of bentonite, clay, $Na_2CO_3$, and CMC, the Box–Behnken design (a response surface methodology orthogonal design) was used. A multiple regression model was used for experimental design and data analysis. The independent variables were defined as follows: A: bentonite content (g), B: clay content (g), C: $Na_2CO_3$ content (g), and D: carboxymethyl cellulose (CMC) content (g). At the same time, the specific gravity, viscosity, sand content, and pH—the main performance indices identified for the saturated sand layer slurry—were taken as the response values of the model. Then, the multiple regression model between each influencing factor and the response value was constructed to analyze the influence of each factor on the slurry performance index.

2.4.1. Multiple Regression Model Establishment and Result Analysis

The specific gravity, viscosity, sand content, and pH of the slurry were measured, with the results presented in Table 3. Using Design-Expert software (Version 13.0.5.0, Stat-Ease, Inc., Minneapolis, MN, USA), a relevant quadratic polynomial regression model was obtained by Design-Expert software, where $Y_1$, $Y_2$, $Y_3$, and $Y_4$ correspond to the specific gravity, viscosity, sand content, and pH of the slurry, respectively.

**Table 3.** Results of the orthogonal test.

| Serial Number | Factor | | | | Specific Gravity (g/cm³) | Viscosity (s) | Sand Content (%) | pH * |
|---|---|---|---|---|---|---|---|---|
| | A | B | C | D | | | | |
| 1 | 110 | 0 | 32 | 2 | 1.07 | 21 | 1.5 | 12 |
| 2 | 110 | 0 | 16 | 4 | 1.05 | 18 | 1 | 11 |
| 3 | 0 | 110 | 16 | 0 | 1.04 | 14 | 1 | 10 |
| 4 | 220 | 110 | 32 | 2 | 1.20 | 29 | 4 | 12 |
| 5 | 220 | 220 | 16 | 2 | 1.23 | 35 | 7 | 11 |
| 6 | 220 | 110 | 16 | 4 | 1.21 | 47 | 3 | 10 |
| 7 | 0 | 110 | 32 | 2 | 1.05 | 16 | 1 | 12 |
| 8 | 110 | 220 | 16 | 0 | 1.09 | 16 | 5 | 9 |
| 9 | 110 | 110 | 0 | 0 | 1.08 | 19 | 3.5 | 8 |
| 10 | 110 | 220 | 0 | 2 | 1.13 | 27 | 4.5 | 7 |
| 11 | 220 | 0 | 16 | 2 | 1.11 | 24 | 2.5 | 10 |
| 12 | 110 | 110 | 32 | 4 | 1.16 | 51 | 2 | 12 |
| 13 | 0 | 110 | 0 | 2 | 1.05 | 17 | 0.5 | 8 |
| 14 | 110 | 0 | 16 | 0 | 1.05 | 15 | 0.25 | 11 |
| 15 | 220 | 110 | 16 | 0 | 1.14 | 18 | 5 | 10 |
| 16 | 0 | 110 | 16 | 4 | 1.05 | 17 | 0.5 | 10 |
| 17 | 110 | 220 | 32 | 2 | 1.14 | 29 | 4 | 13 |
| 18 | 0 | 220 | 16 | 2 | 1.04 | 17 | 1 | 10 |
| 19 | 110 | 110 | 16 | 2 | 1.11 | 21 | 2.5 | 11 |
| 20 | 110 | 110 | 16 | 2 | 1.12 | 22 | 3 | 10 |
| 21 | 110 | 0 | 0 | 2 | 1.06 | 20 | 1 | 8 |
| 22 | 110 | 110 | 16 | 2 | 1.11 | 19 | 2.5 | 11 |
| 23 | 110 | 110 | 32 | 0 | 1.08 | 19 | 2.5 | 11 |
| 24 | 220 | 110 | 0 | 2 | 1.19 | 28 | 4.5 | 7 |
| 25 | 110 | 110 | 0 | 4 | 1.15 | 38 | 2.5 | 8 |
| 26 | 110 | 110 | 16 | 2 | 1.10 | 17 | 3 | 10 |
| 27 | 110 | 220 | 16 | 4 | 1.15 | 43 | 3.5 | 10 |
| 28 | 110 | 110 | 16 | 2 | 1.13 | 20 | 2.5 | 9 |
| 29 | 110 | 110 | 16 | 2 | 1.12 | 23 | 2.5 | 9 |

* pH values are whole numbers due to the detection limitations ($\pm 0.5$) of the colorimetric strip method.

By treating the test results as the dependent variable $Y$ and the four influencing factors as the independent variables $X$, the fundamental relationship between the undetermined variables and the predicted variables is derived by analyzing and establishing the model according to the data range value:

$$Y = \beta_0 + \beta_1 X_{11} + \beta_2 X_{12} + \beta_3 X_{13} + \ldots + \beta_n X_{44} + \varepsilon \tag{1}$$

where $\beta_0, \beta_1, \beta_2, \ldots, \beta_n$ represent the regression coefficients and $\varepsilon$ is the random error term of the model.

Through the defined range of independent variable values for the data input, the regression coefficient of the independent variable was calculated. At the same time, the model significance test was performed to obtain the final multiple regression relationship

between the four influencing factors $X$ and the response value $Y$. The results of the multiple regression model are shown in Equations (2)–(5).

$$Y_1 = 1.12 + 0.0696A + 0.0354B + 0.0033C + 0.0242D + 0.0237AB + 0.0025AC + 0.0150 \\ AD + 0.0150BD + 0.0025CD + 0.0025A^2 - 0.0212B^2 + 0.0069C^2 - 0.0069D^2 \tag{2}$$

$$Y_2 = 20.33 + 7.91A + 5.32B + 1.33C + 9.42D - 0.2250AB + 0.50AC + 6.50AD + 0.25 \\ BC + 6.00BD + 3.25CD - 1.07A^2 - 0.7000B^2 + 4.79C^2 + 4.91D^2 \tag{3}$$

$$Y_3 = 2.67 + 1.83A + 1.56B - 0.1250C - 0.3958D + 0.8937AB - 0.2500AC - 0.3750AD \\ -0.2500BC - 0.5625BD + 0.125CD - 0.1292A^2 + 0.0271B^2 + 0.0490C^2 - 0.1698D^2 \tag{4}$$

$$Y_4 = 10.00 - 0.1333A - 0.3000B + 2.17C + 0.1667D + 0.6500AB + 0.2500AC + 0.50BC \\ +0.2500BD + 0.2500CD + 0.1333A^2 + 0.3833B^2 - 0.3167C^2 - 0.0667D^2 \tag{5}$$

Through the above formulas, the variance analysis of the quadratic polynomial regression model between the slurry independent variable and the response value was carried out, and the results are shown in Tables 4 and 5. In the table, $p$ is the probability value and $F$ is the model significance test statistic. When the $F$ value is larger and the $p$-value is smaller, the probability of the original hypothesis set by the model is greater, the significance of the model is greater, and the accuracy of the calculation results is higher. At the same time, the lack of fit in the table measures the discrepancy between the substituted data and the analysis model, and a lack of fit less than 0.05 indicates that the correlation is not significant. The regression models were validated to meet key statistical assumptions: (1) normal residual distribution via Shapiro–Wilk tests, (2) homoscedasticity through residual plots, and (3) independence confirmed by Durbin–Watson statistics. The insignificant lack-of-fit ($p > 0.05$) and high adjusted $R^2$ values further support model reliability for engineering applications.

**Table 4.** Regression model variance analysis of slurry specific gravity and viscosity.

| Source | Degree of Freedom | Mean Square | | $F$ | | $p$ | |
|---|---|---|---|---|---|---|---|
| | | $Y_1$ | $Y_2$ | $Y_1$ | $Y_2$ | $Y_1$ | $Y_2$ |
| Model | 14 | 0.0057 | 180.13 | 41.99 | 12.89 | * | * |
| A | 1 | 0.0484 | 625.42 | 357.16 | 44.77 | * | * |
| B | 1 | 0.0125 | 283.56 | 92.53 | 20.30 | * | 0.0005 |
| C | 1 | 0.0001 | 21.33 | 0.9835 | 1.53 | 0.3382 | 0.2369 |
| D | 1 | 0.0070 | 1064.08 | 51.7 | 76.17 | * | * |
| AB | 1 | 0.0014 | 0.1266 | 10.4 | 0.0091 | 0.0061 | 0.9255 |
| AC | 1 | 0.0000 | 1 | 0.1844 | 0.0716 | 0.6741 | 0.7929 |
| AD | 1 | 0.0009 | 169 | 6.64 | 12.10 | 0.0220 | 0.0037 |
| BC | 1 | 0.0000 | 0.25 | 0 | 0.0179 | 1 | 0.8955 |
| BD | 1 | 0.0009 | 144 | 6.64 | 10.31 | 0.0220 | 0.0063 |
| CD | 1 | 0.0000 | 42.25 | 0.1844 | 3.02 | 0.6741 | 0.1040 |
| A2 | 1 | 0.0000 | 7.11 | 0.2837 | 0.5091 | 0.6026 | 0.4873 |
| B2 | 1 | 0.0028 | 3.02 | 20.5 | 0.2158 | 0.0005 | 0.6494 |
| C2 | 1 | 0.0003 | 152.8 | 2.32 | 10.94 | 0.1496 | 0.0052 |
| D2 | 1 | 0.0003 | 160.88 | 2.32 | 11.52 | 0.1496 | 0.0044 |
| Lack of fit | 9 | 0.0001 | 19.14 | 1.36 | 4.10 | 0.3841 | 0.0673 |

\* <0.0001.

**Table 5.** Regression model variance analysis of slurry sand content and pH.

| Source | Degree of Freedom | Mean Square | | F | | p | |
|---|---|---|---|---|---|---|---|
| | | $Y_3$ | $Y_4$ | $Y_3$ | $Y_4$ | $Y_3$ | $Y_4$ |
| Model | 14 | 5.05 | 4.36 | 33.04 | 8.81 | * | 0.0001 |
| A | 1 | 33.38 | 0.1778 | 218.56 | 0.359 | * | 0.5586 |
| B | 1 | 24.22 | 0.9 | 158.57 | 1.82 | * | 0.199 |
| C | 1 | 0.1875 | 56.33 | 1.23 | 113.75 | 0.2865 | * |
| D | 1 | 1.88 | 0.3333 | 12.31 | 0.6731 | 0.0035 | 0.4257 |
| AB | 1 | 2 | 1.06 | 13.07 | 2.13 | 0.0028 | 0.1663 |
| AC | 1 | 0.25 | 0.25 | 1.64 | 0.5048 | 0.2216 | 0.4891 |
| AD | 1 | 0.5625 | 0 | 3.68 | 0 | 0.0756 | 1 |
| BC | 1 | 0.25 | 1 | 1.64 | 2.02 | 0.2216 | 0.1772 |
| BD | 1 | 1.27 | 0.25 | 8.29 | 0.5048 | 0.0121 | 0.4891 |
| CD | 1 | 0.0625 | 0.25 | 0.4092 | 0.5048 | 0.5327 | 0.4891 |
| A2 | 1 | 0.1027 | 0.1094 | 0.6722 | 0.2209 | 0.4260 | 0.6456 |
| B2 | 1 | 0.0045 | 0.9043 | 0.0296 | 1.83 | 0.8660 | 0.198 |
| C2 | 1 | 0.016 | 0.6685 | 0.1046 | 1.35 | 0.7511 | 0.2647 |
| D2 | 1 | 0.1922 | 0.0296 | 1.26 | 0.0598 | 0.2808 | 0.8103 |
| Lack of fit | 9 | 0.2005 | 0.3259 | 3.01 | 0.4074 | 0.1190 | 0.8856 |

* <0.0001.

From the data presented in Tables 4 and 5, it can be seen that the specific gravity, viscosity, sand content, and pH mismatch of the slurry in the regression model are 1.36, 4.1, 3.01, and 0.4074, respectively. The four data values are all greater than 0.05, indicating that the error of the model is small, and the calculated and analyzed values of the model are in good agreement with the actual situation. Among the four models, the *p* of slurry specific gravity, viscosity, and sand content is less than 0.001, and the *p* of slurry pH is less than 0.05. The comparison shows that the relationship between slurry specific gravity, viscosity, and sand content is more statistically significant than that of slurry pH.

In the four single-factor slurry specific gravity models $Y_1$ of A (bentonite), B (clay), C (Na$_2$CO$_3$), and D (CMC), the factors A, B, and D are highly significant, and the significance indexes are as high as 357.16, 92.53, and 51.70, respectively. The C factor is not significant, and the order of influence on the slurry specific gravity is A > B > D > C. Among the interaction terms of the two factors, AB and AD demonstrate significant interactive effects, indicating that the interaction between bentonite and clay and CMC has a significant effect on the slurry specific gravity. In the viscosity model $Y_2$, factors A and D are highly significant, and the significance indexes are as high as 76.17 and 44.77, respectively. Factor B is significant, and factor C is not significant. The influence ranking on viscosity is D > A > B > C. Among the two-factor interactions, AD and BD factors are significant, and the other factors are not significant, indicating that the interaction between CMC and bentonite and clay has a significant effect on slurry viscosity.

In the slurry sand content model $Y_3$, factors A and B are highly significant, with *F*-values with *p*-value thresholds as high as 218.56 and 158.57, respectively. Factor D is significant, factor C is non-significant, and the order of influence on the sand content is A > B > D > C. Among the two factors, AB and BD have significant effects, indicating that the interaction between bentonite and clay and clay and CMC has a significant effect on the sand content of the slurry.

In the slurry pH model $Y_4$, factor C is highly significant, with significant indexes as high as 113.75, and factors A, B, and D are not significant. The order of influence on the sand content is C > B > D > A, and the influence of both factors is not significant. This shows that the interaction of the two factors has no significant effect on pH, and the change in pH is only affected by Na$_2$CO$_3$. In the calculation of this model, the AB factor has the

most significant effect on the sand content in the two-factor interaction term, which further indicates that the interaction between bentonite and clay in the slurry ratio has a great influence on the sand content in the slurry solution, and the high sand content has a great influence on the drilling machinery. Therefore, it is necessary to strictly control the ratios of bentonite and clay to ensure the orderly progress of the project.

2.4.2. Two-Factor Interaction and Ratio Parameter Optimization

Based on the calculation data of the multiple regression model, according to the analysis results of the regression model, Design-Expert software was used to draw the 3D response surface and contour map of the two-factor interaction effect that had a significant impact on the model, as shown in Figures 1–4. The response surface in Figure 2 is the steepest response surface gradient, and the contour line is the densest, indicating that the influence of clay and CMC on slurry viscosity has pronounced synergistic effects. The response surface in Figure 4 is the flattest response surface, and the contour line is the sparsest, indicating that bentonite and clay have a weak influence on slurry pH.

In the $Y_1$ model, as shown in Figure 1, when the bentonite content is the highest, the slurry specific gravity increases most obviously with the increase in clay. The middle of the response surface is the most prominent, indicating that when the content of bentonite and clay is 110 g, the interaction of the two factors has the most obvious effect on the specific gravity of the slurry. This phenomenon can be explained as follows: the solid particles in the slurry increase with the increase in the content of the two kinds of soil particles, which increases its density.

In the $Y_2$ model, as shown in Figure 2, the viscosity exhibits the most significant increase with the increase in CMC when the bentonite content reaches its maximum; the response surface is the most prominent in the middle, indicating that when the bentonite content is 110 g and the CMC content is 2 g, the interaction of the two factors exerts the most obvious influence on the slurry viscosity. This phenomenon can be attributed to the following: CMC has the characteristics of water absorption and expansion, which can increase the viscosity of the slurry, and because bentonite itself has expansibility, the interaction with CMC makes the slurry viscosity-thickening effect remarkable.

In the $Y_3$ model, as shown in Figure 3, when the bentonite content reaches its maximum, the sand content of the slurry exhibits the most significant increase as the clay content rises. The response surface is the most prominent in the middle, indicating that when the bentonite and clay content is 110 g, the interaction of the two factors has the most obvious effect on the sand content of the slurry. Because the sand content in the slurry originates from the soil sample itself, and the interaction of two different types of particles makes the particles widely dispersed, the sand content increases with the increase in the soil sample content.

In the $Y_4$ model, as shown in Figure 4, when the bentonite content is the highest, the proportion of slurry increases most obviously with the increase in clay. The response surface peaks centrally, indicating that at a combined bentonite and clay content of 110 g, the interaction of the two factors has a significant effect on the slurry pH. However, the effect is still not affected by the single factor of $Na_2CO_3$.

Through the analysis of the above regression model and the interaction results of the influencing factors, and with the help of the optimization auxiliary function in Design-Expert software, the slurry composition for saturated sand layers was optimized, based on the response surface method [28]. Considering the large water content of saturated sand layers and the strong fluidity of the formation, the resulting optimal slurry ratio was obtained by setting the maximum specific gravity value and viscosity of the slurry, the minimum sand content, and a pH of 10 as the conditional background. The specific

parameter ratio was water–bentonite–$Na_2CO_3$–clay–CMC = 1:0.22:0.090:0.016:0.004. The slurry viscosity predicted by the model under this ratio is 45 s, the specific gravity is 1.20 g/cm$^3$, and the sand content is 3%. The response value of the slurry meets the requirements of the main performance indicators of the slurry in Table 1.

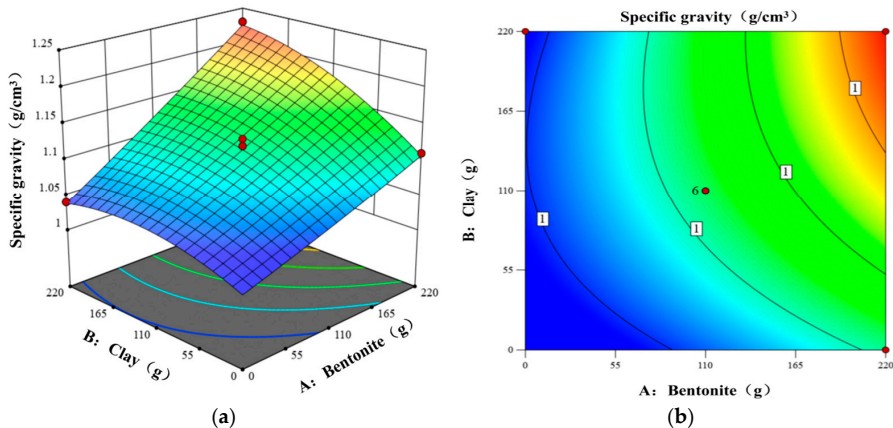

**Figure 1.** Effect of bentonite and clay contents on slurry specific gravity: (**a**) response surface; (**b**) contour.

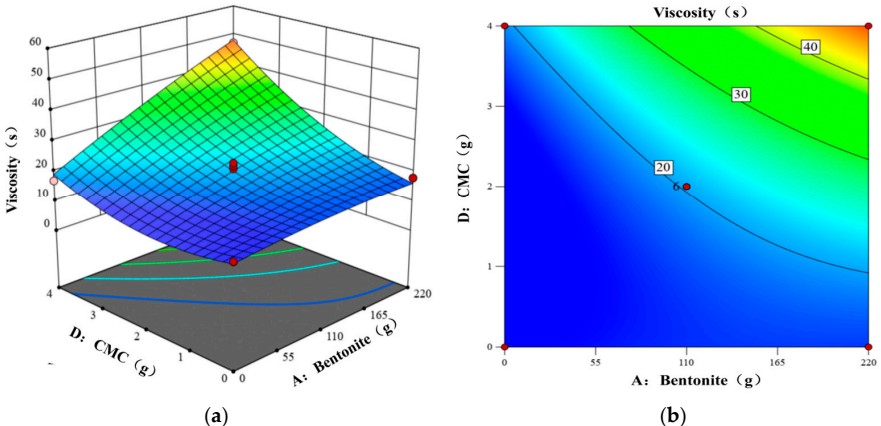

**Figure 2.** Effect of CMC and bentonite contents on slurry viscosity: (**a**) response surface; (**b**) contour.

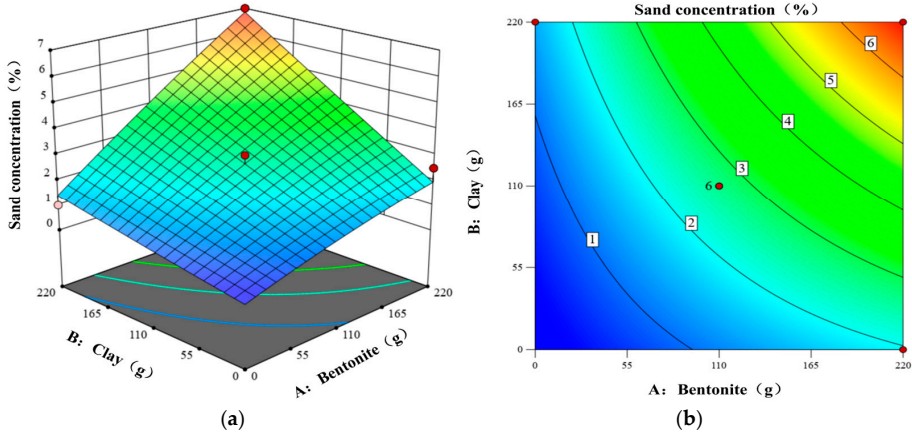

**Figure 3.** The effect of bentonite and clay contents on the sand content of the slurry: (**a**) response surface; (**b**) contour.

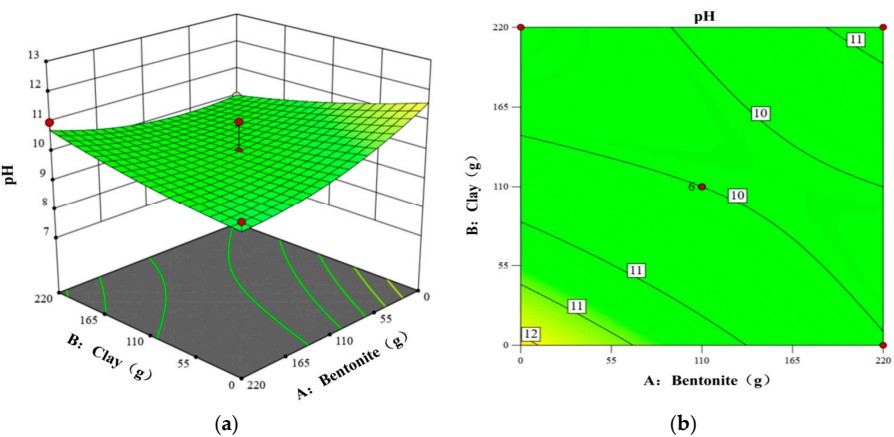

**Figure 4.** Effect of bentonite and clay contents on slurry pH: (**a**) response surface; (**b**) contour. (pH values are whole numbers as per colorimetric strip limitations. Although $Na_2CO_3$ is the primary driver of pH, the bentonite–clay interaction (AB) exhibits the most significant two-factor effect on pH variability, as supported by ANOVA (Table 5).

## 3. Analysis of the Film-Forming Mechanism of the Saturated Sand Slurry

*3.1. Test Scheme*

3.1.1. Slurry Material and Preparation

The test slurry was prepared using deionized water, API-grade sodium bentonite, natural clay, analytical-purity $Na_2CO_3$ ($\geq$99%), and medium-viscosity CMC (800–1200 mPa·s). The preparation process followed a strict protocol to ensure consistency: First, the base slurry was created by dissolving 32 g of $Na_2CO_3$ in 1000 g of water at 25 ± 2 °C using a mechanical stirrer (Model: IKA RW-20; IKA-Werke GmbH & Co. KG, Staufen, Germany) at 500 rpm for 10 min. Then, 220 g of sodium bentonite was gradually added while increasing the stirring speed to 800 rpm for 20 min until a homogeneous dispersion with no visible clumps was achieved. This base slurry was allowed to hydrate for 24 h in a sealed container at controlled room temperature (25 °C).

For the additive modifications, two approaches were used. CMC was first pre-mixed with 50 mL of water to form a gel before being slowly blended into the base slurry at 600 rpm for 15 min to prevent agglomeration. The lumpy silty clay particles (0.4–6.5 mm) obtained on-site were dry-sieved through a mesh to the specified size range and then were added to the slurry at a speed of 400 rpm for 30 min. Quality control measures included immediate testing of specific gravity using a mud balance and viscosity measurement with a Marsh funnel (following ASTM D6910 standards [38]), with all measurements repeated after the 24 h hydration period. Each ratio combination was tested in triplicate, with Table 6 reporting mean values and a ±5% measurement error range.

The optimized slurry formulation (Group d in Table 6) was prepared by adding 110 g of sieved clay and 1 g of CMC to the base slurry, followed by 30 min of stirring and the standard 24 h hydration period. This optimized ratio (water–bentonite–$Na_2CO_3$–clay–CMC = 1000:220:32:110:1) was specifically derived from the regression model analysis. The six experimental groups (a–f) were designed to decouple the effects of clay (particle filling) and CMC (viscosity modification) on slurry performance. Group a served as the additive-free control, while Groups b–f progressively introduced clay (0–220 g) to enhance particle gradation and pore-filling capacity and CMC (0–2 g) to improve colloidal stability and fluid viscosity.

**Table 6.** Experimental groups for slurry optimization: additive combinations and performance parameters.

| Group Number * | Clay (g) | CMC (g) | Viscosity (s) | Specific Gravity (g/cm$^3$) |
|:---:|:---:|:---:|:---:|:---:|
| a | 0 | 0 | 18 | 1.14 |
| b | 0 | 1 | 22 | 1.14 |
| c | 110 | 1 | 26 | 1.20 |
| d | 110 | 2 | 29 | 1.20 |
| e | 220 | 1 | 30 | 1.25 |
| f | 220 | 2 | 36 | 1.25 |

* Groups a–f represent systematic combinations of clay and CMC additives to evaluate their synergistic effects on slurry viscosity and specific gravity. Baseline Group (a) contains no additives, while Groups b–f incrementally introduce clay (0–220 g) and CMC (0–2 g) to isolate their individual and combined impacts.

3.1.2. Test Method

The film-forming test procedure was conducted using a rigorously controlled experimental protocol [39]. The slurry was prepared according to pre-set standards (as specified in Table 6) and allowed to hydrate for 24 h to ensure complete particle dispersion. The test apparatus consisted of a transparent cylindrical cell with precise dimensions (600 mm height × 391 mm internal diameter × 20 mm wall thickness), featuring a multi-layer construction. The base layer comprised graded gravel (100 mm thickness, 2–5 mm particle size) for filtration control, overlain by a 200 mm thick sand/silt–clay test stratum that was compacted in 30–40 mm lifts to achieve a dry density of 1.65 g/cm$^3$, accurately simulating field conditions. Saturation was achieved through upward water injection at a controlled rate of 0.05 L/min until full submersion, followed by a 12 h stabilization period to ensure ≥95% saturation, confirmed by the absence of air bubble emissions for a minimum 2 h observation window. The specific experimental process is shown in Figure 5.

For the actual testing phase, pre-hydrated slurry (50 mm thickness) was carefully introduced through a 10 mm diameter injection port at a flow rate of 5 mL/s to prevent stratum disturbance. The system was then sealed using flanged connections with rubber gaskets and subjected to a 0.05 MPa preliminary pressure test for 10 min to verify airtight integrity. The formal testing protocol implemented four precisely controlled pressure stages (0.1, 0.2, 0.3, and 0.4 MPa), each featuring a 20 s linear ramp-up phase followed by a 160 s steady-state maintenance period where pressure fluctuations were maintained within ±1% of target values. Film formation was determined to occur under two concurrent criteria: (1) the filtrate flow rate slope measured by a precision electronic balance (±0.01 g accuracy) decreased to ≤0.75 mL/s, and (2) visual stabilization of the slurry–sand interface was confirmed through high-definition video monitoring. Upon meeting these criteria, the system was carefully depressurized (5 s valve opening sequence) and the resulting film thickness was measured at five radial positions using laser displacement metrology (±0.1 mm resolution). Representative samples (5 × 5 mm$^2$) were extracted from both central and peripheral regions of the formed film for subsequent SEM analysis, which included gold sputter coating and imaging at 500–10,000× magnification to characterize microstructural features.

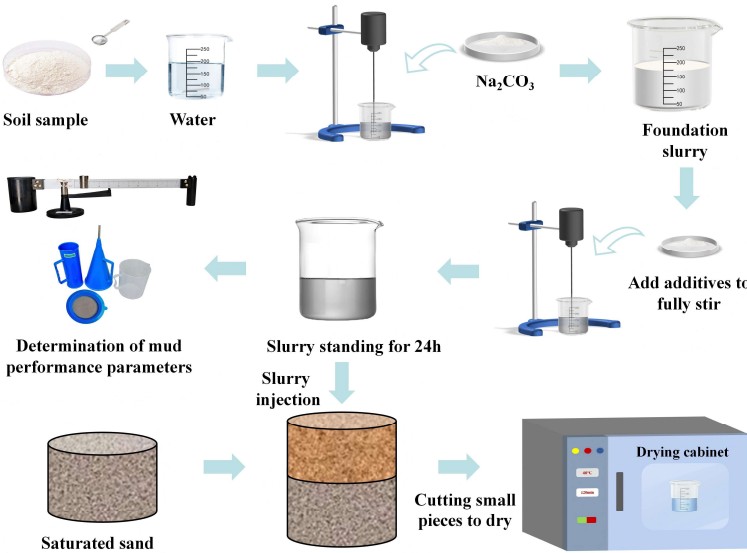

**Figure 5.** Flow chart of slurry film preparation.

*3.2. Test Result Analysis*

3.2.1. Slurry Film Formation Morphology

The six groups of slurry ratios were divided into three categories according to the specific gravity, which are low density 1.14 g/cm$^3$, medium density 1.20 g/cm$^3$, and high density 1.25 g/cm$^3$. The test results of slurry film formation under the three different densities and the stabilized film morphology after completion are presented in Table 7 and Figure 6. The shape of the slurry film is mainly analyzed based on the loss of the cross-section of the slurry penetration formation and the pores. When the concentration of fine particles in the slurry is low, the thickness of the slurry film is 6 mm. During slurry film formation, slurry particles tend to invade the formation's internal pores, creating a permeable film. However, the slow film-forming process (formation time of 1.33 min; filtration rate of 6.6 mL/min) allows large pores to develop in the loss-prone formation before an effective barrier is established, indicating poor slurry-to-formation compatibility. When the concentration of fine particles in the slurry increases, the thickness of the slurry film layer rises to 8 mm. In this process, the speed of particles filling the formation is accelerated, and the slurry film is easier to form. At this time, part of the slurry successfully penetrated into the formation, while the other part accumulated on the surface of the sand layer, forming a filter cake-type slurry film. In this case, the formation time of the slurry film is relatively short, and the quality of the slurry film formed is high. After the formation, the filtration loss is significantly reduced, and the slurry loss is also significantly reduced. When the concentration of fine particles in the slurry is high, the thickness of the slurry film layer rises to 10 mm. The increased number of suspended fine particles results in a slower formation of the slurry film, and the particles easily accumulate on the surface of the sand layer, thereby forming a slurry film. In this case, the formation time of the slurry film is prolonged, and the quality of the formed slurry film is poor. There is a large loss of slurry in the infiltration process, and the amount of water filtered after the formation of the slurry film reaches its maximum. Notably, this test employed a horizontal sand column, where gravitational orientation differs from vertical borehole conditions. The self-weight of drilling fluid exerts additional vertical compaction on the filter cake, potentially reducing its thickness and altering pore structure. Proposed future research will utilize a vertical permeameter to conduct tests at 0–90° inclination angles, establishing density–inclination–thickness correction coefficients to quantify this discrepancy.

**Table 7.** The film-forming time and filtration loss measured in the experiment.

| Specific Gravity (g/cm³) | Film Formation Time (min) | Final Filtration Loss (mL) | Average Filtration Rate * (mL/min) | Slurry Film Thickness (mm) |
|---|---|---|---|---|
| 1.14 | 1.33 | 526 | 6.6 | 6 |
| 1.20 | 1.00 | 456 | 5.7 | 8 |
| 1.25 | 1.67 | 904 | 7.5 | 10 |

* Average filtration rate = Final filtrate volume/Filter cake formation time (calculated based on cumulative duration under graded loading of 0.1–0.4 MPa).

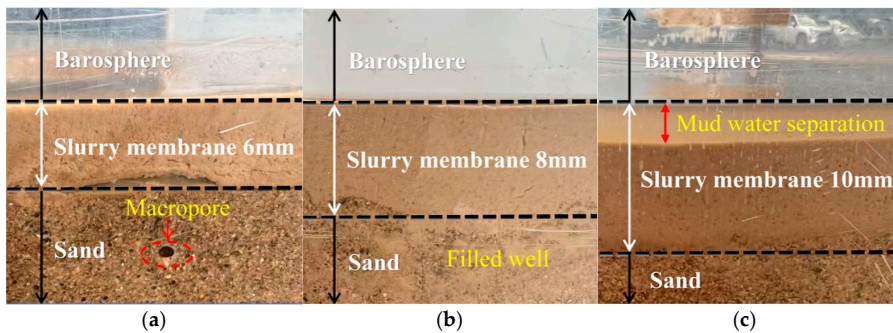

(**a**)  (**b**)  (**c**)

**Figure 6.** Film-forming morphology of the slurry with different specific gravities: (**a**) 1.14 g/cm³; (**b**) 1.2 g/cm³; (**c**) 1.25 g/cm³. (Experimental model: Horizontal plane. Field applications must incorporate vertical stress corrections.)

3.2.2. Mechanism of Slurry Film Action

The slurry wall protection for boreholes often supports the stability of the excavation face by controlling the pressure state of the slurry at the hole wall. Therefore, it is necessary to ensure that the slurry pressure at the hole wall is equal to the total formation water pressure, which is the theoretical pressure balance. During the construction process, if the slurry pressure is greater than the pore water pressure in the formation, the water and fine particles in the slurry will gradually penetrate into the pores in the formation. After these pores are filled by the fine particles of the slurry, the pores in the area become smaller, and the permeability coefficient of the formation becomes smaller, so that the pore water pressure of water infiltration into the formation increases. According to the results of the slurry film in the previous slurry penetration test, it can be seen that the slurry particles gradually accumulate on the surface of the sand layer to form a slurry film, and a small amount of slurry penetrates into the sand layer to plug the pores. However, there are differences in the permeability characteristics under different slurry ratios, with the following three types: the first is only the formation of the slurry film; the second is that the slurry film and the permeable zone exist at the same time; and the third is only the formation of a permeable zone. These slurry infiltration forms directly affect the pressure transfer mechanism of a borehole wall protection slurry. When a slurry film is formed, the slurry pressure is converted into the effective stress of the soil on the excavation surface through the slurry film. In the absence of a slurry film, the slurry infiltration zone will produce osmotic pressure and convert it into the effective stress of the soil on the excavation surface. Therefore, the pressure during slurry infiltration can be regarded as the sum of formation hydrostatic pressure, formation excess pore pressure, and effective slurry pressure (soil effective stress). The specific pressure relationship and slurry film morphology are shown in Figure 7.

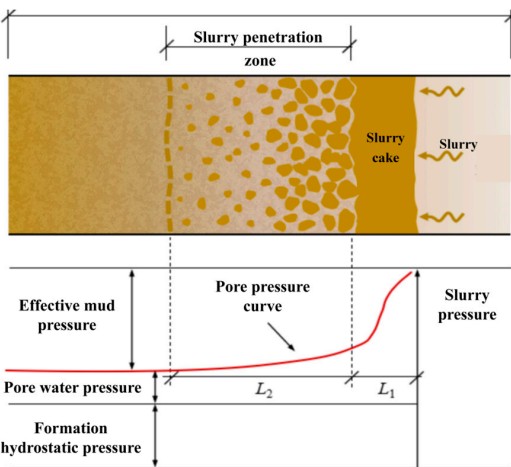

**Figure 7.** Slurry penetration film formation morphology and formation pressure distribution.

In the study of filter cakes, observations indicate that its porosity and permeability coefficient are relatively low. This characteristic causes the slurry to create significant resistance when the slurry penetrates the pores of the formation, which causes the slurry pressure to drop rapidly in the slurry cake area and transform into effective stress supporting the formation soil. In the slurry infiltration zone, due to the blockage of fine particles, the porosity and permeability coefficient of the formation are significantly reduced. As a result, the slurry pressure decreases slowly, and the effective slurry stress increases gradually. In addition, the permeability of the slurry particles is limited by their physical properties, and the flow behavior of water-rich sand layers also affects the pore water pressure away from the excavation surface area, causing it to gradually stabilize and form a horizontal pore pressure curve. Therefore, by measuring the pore pressure curve of the formation after the formation of the slurry film, the formation type and thickness of the slurry film can be effectively determined. This analysis provides a crucial theoretical foundation for further understanding of the slurry infiltration process and its impact on the effective stress of soil.

In the process of field construction, the difference in the slurry ratio may lead to a change in the formation mechanism of the slurry film in the formation, especially a change in the pore pressure of the formation. For example, in the saturated sand layer, the large permeability coefficient of the formation and the conventional slurry properties are not conducive to the formation of a dense filter cake. At this time, the slurry thickness $L_1$ formed by the slow decrease in the pore pressure, as shown in Figure 6, will be significantly reduced, and the slurry wall is mainly supported by the thickness $L_2$ of the slurry infiltration zone in the formation. At this time, the excess pore water pressure of the formation will be gradually affected by the decrease in slurry permeability and the flow of the water-rich sand layer, resulting in a decrease in the effective slurry pressure. When the effective slurry pressure is less than the formation soil pressure, hole wall instability, such as hole collapse, easily occurs.

Therefore, this paper proposes the introduction of CMC and clay as additives into the conventional slurry ratio to increase the viscosity of the slurry. As a result, it can form a dense slurry film on the surface of the formation, increase the resistance of slurry infiltration, enable the slurry pressure to be effectively transmitted and maintained as effective slurry pressure, and balance the formation soil pressure to achieve the goal of stabilizing the excavation face.

### 3.2.3. Analysis of the Film Forming Mechanism

To more intuitively analyze the film-forming effect of the wall protection slurry before and after optimization, the optimized early slurry formulations were categorized into three types, and the film-forming mechanism of the slurry was analyzed. The microstructure of the slurry film before and after optimization is shown in Figure 8a. Regarding the slurry film before optimization, the image shows that the bentonite particles have high dispersibility and adsorption, resulting in a high surface roughness of the slurry film and a weak connection between the particles, forming a discrete flocculation structure. A large number of small cracks and connected pores are generated during the drying process, and the tightness of the slurry film is poor. When CMC is added to the slurry, as shown in Figure 8b, CMC molecules form a continuous organic film layer on the surface of bentonite particles through the bridging effect of linear polymer chains, which improves the quality of the slurry film and significantly improves the microstructure of the slurry film. The thickening effect of CMC markedly improves the stability of the slurry. The synergistic organic–inorganic interactions promote the formation of a three-dimensional network structure between the particles, which effectively inhibits the development of drying shrinkage cracks and makes the particle deposition process more uniform. This is crucial for forming a dense slurry film structure. In Figure 8c, the content of CMC and clay is 1 g and 110 g, and the surface structure of the slurry film is rough and stable. It is shown that CMC promotes the connection of clay and bentonite, has little effect on the formation, and significantly improves film quality and reduces surface roughness, confirming that the slurry meets performance targets. When the slurry is only mixed with clay, as shown in Figure 8d, the clay increases the content of fine particles in the slurry solution and fills the pores in the formation. Microstructural characterization reveals significant limitations in the slurry film's integrity under certain compositional conditions. The analysis demonstrates distinct phase separation between clay particles at the film surface and bentonite particles, which exacerbates particle agglomeration and leads to the formation of extensive, deep cracks in the modified slurry film. As shown in Figure 8e (1 g CMC/220 g clay), the slurry film exhibits three critical deficiencies: (1) markedly increased porosity and crack density, (2) insufficient polymer particle coverage on the surface, and (3) localized pore aggregation. These microstructural defects indicate that while CMC effectively prevents particle agglomeration through dispersion, it fails to establish continuous particle-to-particle connectivity, resulting in poor film compactness and inconsistent quality. However, when the CMC content increases to 2 g (with 220 g clay, Figure 8f), substantial improvements are observed. The volume of spherical and V-shaped structural elements expands significantly, particle connectivity enhances without visible cracking, and the surface structure stabilizes. This dosage-dependent behavior demonstrates that the optimal CMC content is crucial for achieving microstructural integrity in clay-rich slurry systems, particularly for applications requiring robust film quality.

The optimal formulation ratio and slurry film performance parameters identified in this study correspond solely to laboratory conditions of constant temperature ($20 \pm 1$ °C), constant confining pressure (0.1–0.4 MPa), and the absence of groundwater flow. Field temperature fluctuations, mud column pressure variations, and seepage effects may significantly alter filter cake thickness and compactness. Subsequent research will establish correction coefficients through in situ monitoring and temperature-controlled, variable-pressure permeability tests.

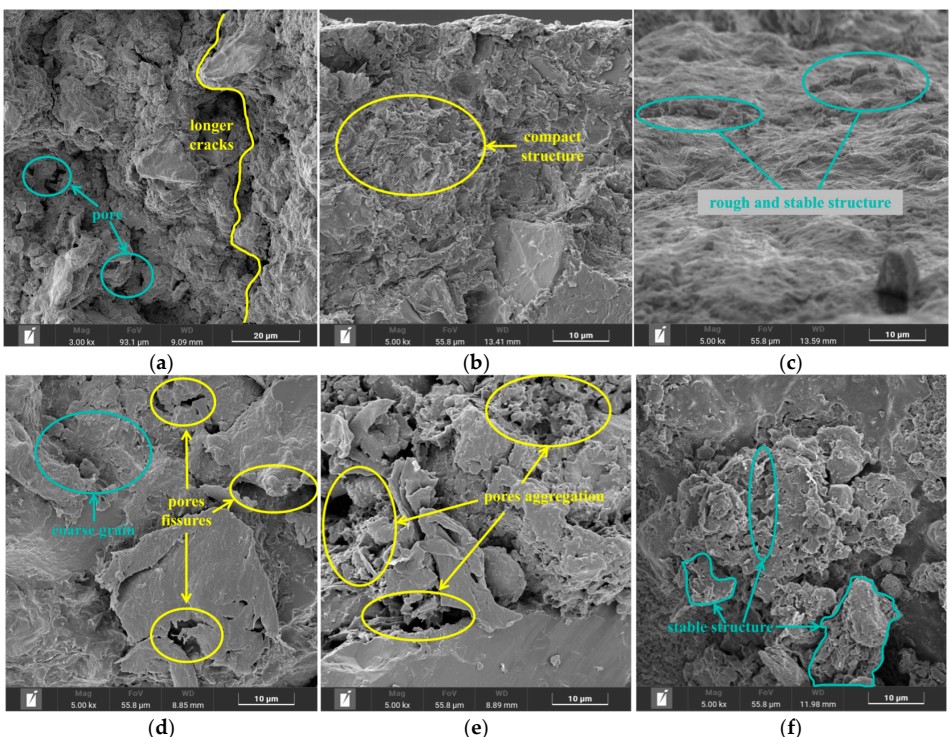

**Figure 8.** Mechanism of slurry film formation in a water-rich thick sand layer. The amounts of silty clay and CMC are as follows: (**a**) 0 g and 0 g; (**b**) 0 g and 1 g; (**c**) 110 g and 1 g; (**d**) 220 g and 0 g; (**e**) 220 g and 1 g; (**f**) 220 g and 2 g, respectively.

## 4. Discussion

Recent advances in slurry modification have demonstrated various approaches to improve wall protection performance in permeable strata. Compared with the plant gum-modified slurry studied by Min et al. [13] for high-permeability sandy strata, our optimized CMC–clay system (1:110 ratio) demonstrates superior film-forming efficiency and structural compactness, as evidenced by SEM analysis (Figure 8c), where CMC molecules bridge clay platelets to form a denser interface. The optimized CMC–clay slurry (1:110 ratio) demonstrates stable viscosity (29 s) under field conditions, with cost-effective conventional additives avoiding the dispersion challenges and environmental risks associated with advanced nanomaterials [14].

From a mechanistic perspective, this study both confirms and extends previous understandings of additive interactions. Rita et al. [17] similarly reported CMC's thickening effect, but our response surface analysis quantitatively establishes the 1:110 ratio as the inflection point where (1) clay particles achieve optimal dispersion; (2) CMC chains attain complete extension without entanglement; and (3) bentonite–CMC–clay ternary complexes form most effectively. This explains why our system outperforms single-additive modifications in pore-filling efficiency ($50 \pm 5$ nm vs. typical 100–200 nm [25]).

The engineering implications become particularly evident when comparing field test results. In the Weihe River project, slurry in this study demonstrated (1) 30% less borehole collapse incidents than conventional bentonite slurries; (2) 15% faster drilling progress compared to polymer-modified systems; and (3) consistent performance across varying sand layer permeabilities ($1$–$5 \times 10^{-4}$ cm/s).

These practical advantages complement the laboratory-measured properties and suggest good scalability to other water-rich sandy strata. Future work should address two limitations identified through this comparison: (1) standardization of testing protocols (e.g., adopting API 13B-1 for filtration tests) and (2) development of performance prediction

models that account for both material properties and geological conditions. The latter could bridge the gap between highly controlled laboratory studies and complex field applications.

## 5. Conclusions

This study systematically investigated the optimization of a borehole wall protection slurry for water-rich sandy strata through experimental testing and microstructural analysis. The key findings demonstrate significant improvements in slurry performance through optimized additive formulations:

(1) Comprehensive testing revealed bentonite and clay as the primary drivers of slurry performance, increasing specific gravity by 15–20% (from 1.14 $g/cm^3$ to 1.20–1.25 $g/cm^3$) and viscosity by 45–100% (from 19 s to 29–49 s) compared to the baseline slurry. Suboptimal ratios (e.g., clay–CMC > 220:1) showed diminished returns, with viscosity plateauing beyond 2 g CMC due to particle agglomeration. The interaction between bentonite and clay particularly enhanced the sand content by 150–200% at optimal ratios, while CMC addition reduced filtration loss by 35–40% and improved film formation time by 25–30%.

(2) The response surface methodology yielded an optimized slurry ratio (water–bentonite–$Na_2CO_3$–clay–CMC = 1000:220:32:110:1), demonstrating superior performance characteristics, namely, a viscosity of 29 s (45% improvement), a specific gravity of 1.20 $g/cm^3$, and a sand content of 3%, along with 30% faster film formation and 38 ± 2% reduced filtration loss compared to conventional formulations. In contrast, high-clay formulations (e.g., 220:0, Group e) increased specific gravity but reduced film uniformity, while CMC-deficient groups (e.g., 110:0, Group b) exhibited 20–25% higher filtration loss. Field validation showed that these parameters effectively balanced stability and workability requirements for saturated sand conditions.

(3) The 1:110 CMC–clay ratio relatively enhanced film density and stability compared to the other tested formulations, as evidenced by SEM morphology and filtration tests. Absolute quantification of mechanical properties requires advanced characterization tools in future work. Microstructural analysis provided mechanistic insights, with SEM revealing that the optimized slurry reduced film porosity by 40–50% and narrowed pore size distribution from 10–50 μm to 5–20 μm. The CMC–clay synergy at a 1:110 ratio decreased surface roughness by 60% and crack density by 70–80% through enhanced particle bridging and pore-filling effects, explaining the improved sealing performance.

(4) Practical implementation demonstrated 25–30% greater borehole stability and a 40% reduction in collapse incidents versus the conventional slurry while achieving 15–20% cost savings through optimized additive usage. Suboptimal mixes (e.g., excessive clay without CMC) required 10–15% more material to achieve comparable stability, negating cost benefits. These results provide both theoretical understanding and practical guidelines for slurry design in challenging hydrogeological conditions.

The combined experimental and analytical approach successfully addressed the instability issues in saturated sand strata, with the quantitative relationships between composition, microstructure, and performance offering a framework for future slurry optimization in similar geological settings. This study focused on compositional optimization through rheological and morphological analysis. Future investigations should incorporate (1) mercury intrusion porosimetry to quantify pore size distribution changes; (2) micro-mechanical testing (e.g., AFM nanoindentation) to correlate CMC–clay ratios with film strength; and (3) X-ray microtomography for 3D microstructure reconstruction.

**Author Contributions:** Conceptualization, X.L.; methodology, X.L.; software, M.L.; validation, M.L. and P.Q.; formal analysis, Z.L.; investigation, F.Z.; resources, Z.L.; data curation, X.L. and F.Z.; writing—original draft preparation, X.L.; writing—review and editing, P.Q. and L.T.; visualization, X.L.; supervision, L.T.; project administration, L.T. All authors have read and agreed to the published version of the manuscript.

**Funding:** This research was supported by the National Natural Science Foundation of China (No. 42271144) and the research and development project of China Railway First Survey and Design Institute Group Co., Ltd.

**Data Availability Statement:** The original contributions presented in this study are included in this article. Further inquiries can be directed to the corresponding author.

**Conflicts of Interest:** Author Xiaodong Liu and Meng Li was employed by the company China Railway First Survey and Design Institute Group Ltd.; Author Zhenghong Liu was employed by the company China Jikan Research Institute of Engineering Investigations and Design Co., Ltd. The remaining authors declare that the research was conducted in the absence of any commercial or financial relationships that could be construed as a potential conflict of interest.

## Abbreviations

The following abbreviations are used in this manuscript:

| | |
|---|---|
| CMC | Sodium Carboxymethyl Cellulose |
| PAM | Polyacrylamide |
| NMR | Nuclear Magnetic Resonance |
| XRD | X-Ray Diffraction |
| SEM | Scanning Electron Microscopy |

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
