# Peer review of "Optimization of the Borehole Wall Protection Slurry Ratio and Film-Forming Mechanism in Water-Rich Sandy Strata"

_2673-4117, doi:10.3390/eng6100251_

Round 1

Reviewer 1 Report

Comments and Suggestions for Authors

Review report for the manuscript titled:

Optimization of Borehole Wall Protection Slurry Ratio and Film-Forming Mechanism in Water-Rich Sandy Stratum

This study examines a significant scientific contribution to the field of geotechnical materials engineering by adopting an advanced methodology to improve the performance of wall protection clay. The paper goes beyond usual methods by using advanced analysis tools like the Response Surface Methodology (RSM) and Scanning Electron Microscopy (SEM) to not only find the best mixing ratio but also show how the protective layer (film) forms at the microscopic level.  The following sections provide some observations, comments, and recommendations that, if taken into consideration, could further enhance the research.

1- Introduction:

The introduction is well-written and successfully defines the main engineering problem, reviews the relevant literature, and clarifies the objectives of the study. It lays a solid foundation for the rest of the paper. However, there are some aspects that could be improved to enhance the strength and impact of the introduction.

  • The phrase "in China" (line 33) makes the problem seem very local, while it is a global issue. It is preferable to use a broader phrasing.
  • In some parts, the literature review reads like a "list of abstracts" (Author A did this, Author B did that). For example, from lines 72 to 86, a series of studies are listed without connecting them together or constructing a coherent argument. My recommendation is instead of simply listing the results, try to synthesize the studies.
  • The paper mentions the use of "clay as an additive" (line 116) as part of its approach. However, clay is a key component, not just an "additive." The real innovation, as is clear from the abstract, lies in improving the ratio between CMC and clay and clarifying how they work together. This crucial point is not sufficiently highlighted in the introduction. This innovative aspect should be strongly highlighted.

2- Optimization scheme of slurry for wall protection of saturated sand layer

This section effectively transitions from the general problem to the specific experimental methodology. It successfully presents the project context, defines performance indicators, explains the function of each material, and, most importantly, rationally justifies the choice of additives. The overall structure is good, but there are some opportunities to improve clarity, conciseness, and scientific impact.

  • Sections 2.1.2 (Basic Properties of Clay) and 2.2.1 (Clay Performance Index) contain a lot of fundamental information like what is taught in university textbooks (for example, an explanation of the four functions of clay, and the definitions of viscosity and density). The target reader of this paper (a researcher or specialized engineer) is expected to already know this information. It is advised to significantly merge and shorten these two sections. Instead of explaining each concept in detail.
  • Table 1 presents excellent target ranges (such as viscosity 20-45 seconds). However, the source of these ranges has not been clarified. Are they from the mentioned codes, from site experience, or from previous studies?
  • The authors state that the sand content with PAM is '1% higher' than with CMC (lines 243-244), a key justification for choosing CMC. However, this claim is ambiguous. The authors should clarify whether this represents a difference of one percentage point (e.g., 4% vs. 3%) or a one percent relative increase (e.g., 3.03% vs. 3%).

3- Analysis of film-forming mechanism of saturated sand slurry

This part provides the essential empirical evidence to support the main hypothesis. The analysis using scanning electron microscopy (SEM) is particularly convincing and provides a valuable contribution. However, there are some serious errors and issues that need significant clarification, which must be addressed to greatly improve the quality and credibility of the paper.

  • On line 396, the author states that the clay particle size used ranges from 0.4 to 6.5 mm. This is a serious error. Clay, by geotechnical definition, consists of particles less than 0.002 mm in diameter. The size mentioned (up to 6.5 mm) falls within the range of coarse sand and fine gravel.
  • There is a lot of confusion in the numbering of the figures. There are two different figures labeled "Figure 5". In addition, in the caption for the figure displaying the images (which is incorrectly numbered as 5), the letter "(a)" was repeated three times instead of using (a), (b), (c). The authors should review all figures and correct the numbering and labels in the text and captions.
  • Section 3.2.1 (Slurry film formation morphology) describes the results qualitatively (for example, "long formation time," "high film quality," "significant reduction in loss"). These claims would be much stronger if supported by numbers. The authors should add a table or mention the numerical values of the data that were likely collected during the experiment, such as: layer formation time (in minutes), and filtration loss rate (ml/min).

Author Response

Comment 1: 

The phrase "in China" (line 33) makes the problem seem very local, while it is a global issue. It is preferable to use a broader phrasing.

Response 1:

We sincerely appreciate the reviewer’s constructive comment. As suggested, we have revised the phrase "in China" (line 35) to a broader phrasing, now stating:

Globally, bored pile technology is widely used for bridge foundation construction.

Comment 2: 

In some parts, the literature review reads like a "list of abstracts" (Author A did this, Author B did that). For example, from lines 72 to 86, a series of studies are listed without connecting them together or constructing a coherent argument. My recommendation is instead of simply listing the results, try to synthesize the studies.

Response 2:

Thank you for this comment. We have revised and polished the literature review section in lines 78-96 of the manuscript, transforming it from a simple enumeration to a categorized synthesis organized by themes, while incorporating critical analysis and discussion of research gaps. The revised content reads as follows:

In the realm of nano-material applications, Edalatfar et al. [19] demonstrated that nano-additives significantly reduce slurry filtration loss by pore-filling effects, though their long-term stability under field conditions remains underexplored. Similarly, Yang et al. [20] achieved a 1.4-fold increase in bentonite swelling index using sodium hexametaphosphate, highlighting the potential of chemical modifiers—yet their cost-effectiveness for large-scale projects warrants further investigation. The shift toward sustainable additives has yielded promising alternatives. Al-Hameedi et al. [21] investigated the impact of incorporating 0.5% to 3.5% black sunflower seed shell powder into drilling mud, demonstrating that this additive across various particle sizes effectively modifies viscosity, controls filtration loss, and enhances wellbore stabilization. Bagum et al. [22] identified Aloe Vera as a non-toxic substitute for conventional chemicals. Lei et al. [2] further expanded this trend by proving agar/guar gum’s superior film-forming ability in sandy strata. However, these studies predominantly focus on laboratory-scale performance, leaving scalability and field compatibility gaps unresolved. For performance optimization, AlAwad et al. [23] standardized cement-rock bonding tests, emphasizing the need for region-specific slurry designs, whereas Nashed et al. [24] pioneered machine learning for bottom-hole pressure prediction—a methodological leap that nonetheless requires validation across diverse geological settings.

Comment 3: 

The paper mentions the use of "clay as an additive" (line 116) as part of its approach. However, clay is a key component, not just an "additive." The real innovation, as is clear from the abstract, lies in improving the ratio between CMC and clay and clarifying how they work together. This crucial point is not sufficiently highlighted in the introduction. This innovative aspect should be strongly highlighted.

Response 3:

We sincerely appreciate this insightful observation. As suggested, we have made the following revisions to better highlight the study's innovative core:

  1. Clarified that clay serves as a base component (not merely an additive).Specific modifications (Lines 110-111):

clay as the fundamental solid-phase material.

  1. Revision in the concluding paragraph of Introduction (Lines 117-122).The following content has been added to emphasize the innovative aspects:

The key innovation of this study involves not simply using conventional clay components, but systematically optimizing their synergistic interaction with CMC. Rigorous ratio control (clay:CMC = 110:1) combined with SEM-based mechanistic analysis reveals how this specific combination significantly enhances film density and pore-filling efficiency, thereby addressing persistent challenges in saturated sand stratum applications.

Comment 4: 

Sections 2.1.2 (Basic Properties of Clay) and 2.2.1 (Clay Performance Index) contain a lot of fundamental information like what is taught in university textbooks (for example, an explanation of the four functions of clay, and the definitions of viscosity and density). The target reader of this paper (a researcher or specialized engineer) is expected to already know this information. It is advised to significantly merge and shorten these two sections. Instead of explaining each concept in detail.

Response 4:

We fully agree with the reviewer's comments and have accordingly merged and streamlined the content in Sections 2.1.2 (in lines 152-157 in the manuscript) and 2.2.1 (in lines 170-192 in the manuscript). The revised concise version reads as follows:

In drilling construction, slurry serves four critical functions: (1) stabilizing the borehole wall by forming a low-permeability slurry film, (2) balancing formation pressure to prevent collapse, (3) lubricating drilling tools to reduce wear, and (4) removing cuttings to maintain hole cleanliness. These functions are optimized through tailored slurry ratios, particularly in saturated sand strata where conventional slurries underperform due to high permeability and weak cementation.

Specific gravity: Controls solid particle concentration to balance formation pressure. Low values risk stratification and poor film formation, while high values increase tool wear and reduce drilling efficiency. Measured using a lever-action mud balance (Model: XYZ-100) following API RP 13B-1. The procedure included horizontal calibration, steady filling to the datum line, and sealing to eliminate surface tension. Readings were taken at torque equilibrium, with triplicate measurements (mean deviation < 1%).

Viscosity: Determines shear resistance and film-forming capacity, essential for mitigating hole collapse and particle settlement in permeable strata. The viscosity of the slurry mentioned in this test was determined using the Marsh funnel viscometer, which measures the time (in seconds, s) required for 500 mL of slurry to flow through the funnel. This is a widely adopted empirical method in drilling engineering for quality control. The apparent viscosity of the slurry was determined to be 15 mPa∙s using a Fann 35 rheometer at 300 rpm, which correlates with a Marsh funnel efflux time of 29 s for 500 mL of slurry.

Sand content: Excessive sand (> 4%) compromises lubrication and equipment longevity, whereas insufficient content reduces cuttings transport. Quantified via sieve analysis (Model: NA-1 sand content set) per ASTM D4380, involving iterative washing/filtration (0.074 mm sieve) and sedimentation to isolate particles > 74 μm.

pH: An alkaline environment optimizes bentonite dispersion and slurry stability while inhibiting bacterial growth. Determined colorimetrically using MColorpHast strips (ISO 3071-compliant) with a resolution of ± 0.5 pH. Strips were immersed in homogenized slurry, and the stabilized color was matched to a reference card. pH was recorded as integers due to the ± 0.5 pH resolution of colorimetric strips.

Comment 5: 

Table 1 presents excellent target ranges (such as viscosity 20-45 seconds). However, the source of these ranges has not been clarified. Are they from the mentioned codes, from site experience, or from previous studies?

Response 5:

  1. Drilling fluid specific gravity: 1.14–1.25 g/cm3

Reference:

[28]Ministry of Geology and Mineral Resources of China. Construction Specifications for Drilled Cast-in-Place Piles (DZ/T 0155-95). Beijing, 1995 (in Chinese).

"In strata prone to borehole collapse, the specific gravity of drilling fluid should be controlled within 1.10–1.30 g/cm³." Based on this specification, the present study refines the range to 1.14–1.25 g/cm³ to maintain borehole wall stability in water-rich thick sand layers while ensuring construction efficiency.

  1. Drilling fluid viscosity:20–45s (this study adopts 18–34 s)
    We sincerely appreciate the reviewer’s valuable suggestions. A new footnote has been added below Table 1 in the revised manuscript to clarify the source of the viscosity range(20–45s):
    "The viscosity range of 20–45s is directly based on specifications for sandy soil strata in the ‘Technical Specifications for Bored Pile Construction’(DZ/T 0155-1995,Article 6.3.2) and ‘Technical Standard for Bored Pile Construction’ (T/CECS 592-2019,Article 5.2.3),with minor adjustments informed by field experience to ensure adaptability in saturated sand layers."

References:

[28]Ministry of Geology and Mineral Resources of China. Construction Specifications for Drilled Cast-in-Place Piles (DZ/T 0155-95). Beijing, 1995 (in Chinese).

[30]Wuhan Survey & Research Institute Co., Ltd. of MCC Group; Hebei Construction Geotechnical Investigation & Research Institute Co., Ltd. Technical Standard for Construction of Bored Pile. China Association for Engineering Construction Standardization (T/CECS 592-2019). Beijing, China, 2019 (in Chinese).

  1. Sand content control threshold: <4%
    Reference:

[31]Wang, H. Study on Preparation and Permeability Performance of Slurry for Bored Piles in Calcareous Cemented Strata. Shenyang University of Technology: Shenyang, China, 2021.

The field monitoring conclusions of this study state:
All formulations in the present investigation rigorously adhere to this threshold. The aforementioned literature has been cited in the corresponding footnote of Table 1 (in lines 168-169) to ensure traceability of parameter sources.

Comment 6: 

The authors state that the sand content with PAM is '1% higher' than with CMC (lines 243-244), a key justification for choosing CMC. However, this claim is ambiguous. The authors should clarify whether this represents a difference of one percentage point (e.g., 4% vs. 3%) or a one percent relative increase (e.g., 3.03% vs. 3%).

Response 6:

We appreciate your insightful observation regarding the ambiguity in the phrase “1% higher sand content”. Upon verification, this 1% represents an absolute difference (i.e.,the PAM group exhibits a value 1 percentage point higher than the CMC group,e.g.,4% vs.3%). To eliminate ambiguity, we have explicitly revised the statement in the manuscript as follows (in lines 241-245):

Under identical additive concentrations, the sand content of PAM-modified slurries was significantly higher compared to CMC-modified slurries, with the absolute difference consistently maintained at approximately 1 percentage point. This phenomenon originates from the coarsening of solid particles induced by PAM’s flocculation effect, as evidenced by the measured data in Table 2.

Comment 7: 

On line 396, the author states that the clay particle size used ranges from 0.4 to 6.5 mm. This is a serious error. Clay, by geotechnical definition, consists of particles less than 0.002 mm in diameter. The size mentioned (up to 6.5 mm) falls within the range of coarse sand and fine gravel.

Response 7:

We appreciate your astute observation regarding the misnomer “0.4–6.5 mm clay”. The original description was indeed inaccurate. We hereby provide a comprehensive clarification of the field sampling and laboratory preparation process:

The 0.4–6.5 mm designation refers to the particle size range of silty clay clods obtained directly from the field, rather than representing the clay fraction. To ensure thorough reaction with bentonite and CMC additives, field soil samples were first pulverized using a crusher and subsequently sieved through a 0.075-mm mesh. This process yielded silty clay particles with a final particle size distribution below 0.075 mm.

A more accurate statement was made in lines 433-435 of the manuscript, as follows:

The lumpy silty clay particles (0.4–6.5 mm) obtained on-site were dry-sieved through a mesh to the specified size range, and then were added to the slurry at a speed of 400 rpm for 30 min.

Comment 8: 

There is a lot of confusion in the numbering of the figures. There are two different figures labeled "Figure 5". In addition, in the caption for the figure displaying the images (which is incorrectly numbered as 5), the letter "(a)" was repeated three times instead of using (a), (b), (c). The authors should review all figures and correct the numbering and labels in the text and captions.

Response 8:

We sincerely appreciate your careful review and valuable feedback. We have thoroughly checked all figures in the manuscript and made the following corrections:

  1. Figure Numbering:

The duplicate "Figure 5" has been renumbered. The slurry film morphology images (originally labeled as two separate "Figure 5") are now correctly labeled as Figure 6.

  1. Subfigure Labels:

The repeated "(a)" in the caption of Figure 6 (previously mislabeled as Figure 5) has been corrected to "(a), (b), (c)" to match the three subfigures.

  1. Cross-verification:

We have double-checked all figure citations and ensured alignment between labels, captions, and in-text references.

Comment 9: 

Section 3.2.1 (Slurry film formation morphology) describes the results qualitatively (for example, "long formation time," "high film quality," "significant reduction in loss"). These claims would be much stronger if supported by numbers. The authors should add a table or mention the numerical values of the data that were likely collected during the experiment, such as: layer formation time (in minutes), and filtration loss rate (ml/min).

Response 9:

We sincerely appreciate the reviewer's valuable suggestions. In the revised manuscript, we have supplemented the Table 7 (in line 522) for clarification and now provide the corresponding quantitative results:

*Average filtration rate = Final filtrate volume / Filter cake formation time (calculated based on cumulative duration under graded loading of 0.1–0.4 MPa)
Revision location: Section 3.2.1. A table has been inserted before Figure 5, and qualitative expressions such as "long formation time" in the original text have been explained with quantitative data: "formation time 1.33 min, filtration rate 6.6 mL/min".

Reviewer 2 Report

Comments and Suggestions for Authors

This manuscript presents a highly relevant and meaningful study that aims to stabilize borehole walls in saturated sandy soils by optimizing slurry composition and elucidating the membrane formation mechanism. The authors' statistical analysis of interactions among multiple additives (bentonite, clay, Na₂CO₃, and CMC) using multiple regression and Box–Behnken design is a significant improvement over conventional single-factor approaches. Additionally, the microscopic evaluation of slurry membranes via scanning electron microscopy (SEM), particularly with regard to porosity and crack development, adds novel insights and enhances our understanding of the functional performance of slurry films.

However, to improve the clarity and impact of the manuscript, I suggest addressing the following points. Section 3.2.3, in particular, contains a large amount of information in lengthy sentences, which compromises readability. Improving visual organization and linking key findings more clearly with figures will help readers better grasp the conclusions. Additionally, the discussion of potential limitations, such as applicability to varying geological conditions, long-term stability, and cost-effectiveness, is limited and should be expanded to clarify the scope of real-world applicability. Several minor English language errors were also noted. I strongly recommend professional English editing to standardize technical terminology and improve sentence clarity and structure.

Below are specific suggestions for revision:

(1) Abbreviations
Before the References section, please provide a dedicated section for abbreviations (e.g., CMC, PAM, BSSSP, NMR, XRD, SEM, and HTP). For formatting standards, refer to MDPI guidelines or previously published papers.

(2) English Editing
There are multiple issues with language usage—for example, the long sentence in Lines 34–38, the inconsistent use of terms such as “slurry film” vs. “slurry skin,” and clear errors in Line 194–195. A comprehensive professional English proofreading is strongly advised.

(3) Viscosity Units
The viscosity is listed in "s," which is ambiguous. Is this meant to refer to Stokes or time in seconds? Please revise the unit to SI standard (Pa·s) to avoid confusion.

(4) Table 1
This appears to list the performance requirements for slurry. Please cite appropriate references to support the values provided.

(5) Lines 207–226
The materials used in the experiments (bentonite, clay, Na₂CO₃, CMC, PAM) should be accompanied by detailed information on the manufacturer and product numbers.

(6) Table 3
Density, viscosity, sand content, and pH were measured, but the measurement methods are not described. Please provide the standards or procedures followed, as well as the instruments used. The pH values are given as integers, which is unusual. As pH differences of 1 correspond to a tenfold change in hydrogen ion concentration, more precise values (e.g., to the second decimal place) are generally expected. If decimal data were recorded, please report them. If your instrument only provides integer readings, clarify this in the caption.

(7) Tables 4 and 5
The sensitivity of response variables is evaluated using F-values. Please confirm whether this is a valid method in this context. If there are precedent studies using this approach, cite them. As a personal comment: I have rarely seen sensitivity assessed by examining the F-values of individual interaction terms (e.g., A, B, AB, AC). I think general method would be to standardize all variables (e.g., via Z-scores) and evaluate the regression coefficients directly.

(8) Figure 3
Is "Silt concentration" a typo for "Sand concentration"?

(9) Figure 4
This figure plots bentonite and clay, but Na₂CO₃ and clay are more relevant to pH changes. Shouldn’t this be plotted instead?

(10) Subsection 3.2.1
The slurry membrane was formed under a horizontally layered condition, but in reality, the borehole membrane forms vertically. Thus, the direction of gravity in Figure 5 (actually Figure 6) does not reflect actual field conditions. The conclusion that membrane thickness increases with slurry density may not hold in the field. Please discuss the differences between the test setup and real-world conditions, and how these may affect membrane formation and thickness.

(11) Line 448–449
Correct:
“Figure 5 → Figure 6”
“(a) 1.14 g/cm³; (a) 1.24 g/cm³; (a) 1.25 g/cm³” →
“(a) 1.14 g/cm³; (b) 1.24 g/cm³; (c) 1.25 g/cm³”

(12) Line 497
Correct “Figure 6 → Figure 7”

(13) Figure 8
Please explain the differences among subfigures (a)–(f) in the caption. According to MDPI guidelines, figure captions should be self-contained and allow the reader to understand the figure without referring to the main text.

Comments on the Quality of English Language

There are multiple issues with language usage—for example, the long sentence in Lines 34–38, the inconsistent use of terms such as “slurry film” vs. “slurry skin,” and clear errors in Line 194–195. A comprehensive professional English proofreading is strongly advised.

Author Response

Comment 1: 

Abbreviations: Before the References section, please provide a dedicated section for abbreviations (e.g., CMC, PAM, BSSSP, NMR, XRD, SEM, and HTP). For formatting standards, refer to MDPI guidelines or previously published papers.

Response 1:

Thank you for your suggestion. We have added a dedicated "Abbreviations" section before the References (Lines 726-728) following MDPI formatting standards. The list includes all technical abbreviations used in the manuscript, with their full names and brief explanations where necessary.

Comment 2: 

English Editing: There are multiple issues with language usage—for example, the long sentence in Lines 34–38, the inconsistent use of terms such as “slurry film” vs. “slurry skin,” and clear errors in Line 194–195. A comprehensive professional English proofreading is strongly advised.

Response 2:

Based on the grammar check performed by specialized software, we have further engaged a native English-speaking editor for comprehensive academic polishing of the manuscript. All revised sections have been highlighted in red throughout the text. The specific modifications to the questions raised by the reviewer are as follows:

  1. Lines 35-38:

Globally, bored pile technology is widely used for bridge foundation construction. However, in saturated sand layers with weak cementation, this method often leads to borehole instability issues such as wall shrinkage and collapse [1], significantly impacting construction schedules and long-term structural performance.

  1. Unify the term slurry skinin the text as slurry film (Line 153 ).
  2. The content of lines 184-187 has been modified and streamlined, and the revised content is as follows:

Sand content: Excessive sand (> 4%) compromises lubrication and equipment longevity, whereas insufficient content reduces cuttings transport. Quantified via sieve analysis (Model: NA-1 sand content set) per ASTM D4380, involving iterative washing/filtration (0.074 mm sieve) and sedimentation to isolate particles > 74 μm.

Comment 3: 

Viscosity Units: The viscosity is listed in "s," which is ambiguous. Is this meant to refer to Stokes or time in seconds? Please revise the unit to SI standard (Pa·s) to avoid confusion.

Response 3:

Thank you for raising this important point regarding viscosity units. We appreciate the opportunity to clarify:

  1. Measurement Method:

The viscosity values were obtained using the Marsh funnel viscometer, which measures the time (in seconds, s) required for 500 mL of slurry to flow through the funnel. This is a widely adopted empirical method in drilling engineering for quality control.

  1. SI Units Clarification:

The “s” unit indeed represents time in seconds, not Stokes.

While Marsh funnel viscosity (s) correlates with apparent viscosity, it does not directly equate to SI units (Pa·s).

3.Revision Made:

To avoid ambiguity, we have supplemented the description regarding viscosity at the specified location in the manuscript (Line 176-183) where thviscosity of slurry is discussed, as follows:

The viscosity of the slurry mentioned in this test was determined using the Marsh funnel viscometer, which measures the time (in seconds, s) required for 500 mL of slurry to flow through the funnel. This is a widely adopted empirical method in drilling engineering for quality control. The apparent viscosity of the slurry was determined to be 15 mPa·s using a Fann 35 rheometer at 300 rpm, which correlates with a Marsh funnel efflux time of 29 s for 500 mL of slurry.

Comment 4: 

Table 1: This appears to list the performance requirements for slurry. Please cite appropriate references to support the values provided.

Response 4:

  1. Drilling fluid specific gravity: 1.14–1.25 g/cm3

Reference:

[28]Ministry of Geology and Mineral Resources of China. Construction Specifications for Drilled Cast-in-Place Piles (DZ/T 0155-95). Beijing, 1995 (in Chinese).

"In strata prone to borehole collapse, the specific gravity of drilling fluid should be controlled within 1.10–1.30 g/cm³." Based on this specification, the present study refines the range to 1.14–1.25 g/cm³ to maintain borehole wall stability in water-rich thick sand layers while ensuring construction efficiency.

  1. Drilling fluid viscosity:20–45s (this study adopts 18–34 s)
    We sincerely appreciate the reviewer’s valuable suggestions. A new footnote has been added below Table 1 in the revised manuscript to clarify the source of the viscosity range(20–45s):
    "The viscosity range of 20–45s is directly based on specifications for sandy soil strata in the ‘Technical Specifications for Bored Pile Construction’(DZ/T 0155-1995,Article 6.3.2) and ‘Technical Standard for Bored Pile Construction’ (T/CECS 592-2019,Article 5.2.3),with minor adjustments informed by field experience to ensure adaptability in saturated sand layers."

References:

[28]Ministry of Geology and Mineral Resources of China. Construction Specifications for Drilled Cast-in-Place Piles (DZ/T 0155-95). Beijing, 1995 (in Chinese).

[30]Wuhan Survey & Research Institute Co., Ltd. of MCC Group; Hebei Construction Geotechnical Investigation & Research Institute Co., Ltd. Technical Standard for Construction of Bored Pile. China Association for Engineering Construction Standardization (T/CECS 592-2019). Beijing, China, 2019 (in Chinese).

  1. Sand content control threshold: <4%
    Reference:

[31]Wang, H. Study on Preparation and Permeability Performance of Slurry for Bored Piles in Calcareous Cemented Strata. Shenyang University of Technology: Shenyang, China, 2021.

The field monitoring conclusions of this study state:
All formulations in the present investigation rigorously adhere to this threshold. The aforementioned literature has been cited in the corresponding footnote of Table 1 ensure traceability of parameter sources.

Comment 5: 

Lines 207–226: The materials used in the experiments (bentonite, clay, Na₂CO₃, CMC, PAM) should be accompanied by detailed information on the manufacturer and product numbers.Ø

Response 5:

We sincerely appreciate your valuable suggestions. Regarding the manufacturer information of experimental materials, we have supplemented and enhanced the relevant details in lines 199-222 of manuscript, as follows:

  • Bentonite: Sourced from Zhejiang Hongyu New Materials Co., Ltd., bentonite serves as the primary solid-phase material in the slurry system. Its high hygroscopicity and adsorption capacity enable it to absorb water and various inorganic substances, forming colloidal particles that seal gaps and fractures to prevent fluid loss. Additionally, bentonite increases slurry density and viscosity, thereby aiding in pore pressure control and borehole wall stabilization. As an economical and environmentally benign material, it offers both technical and operational advantages.

(2) Clay: Collected from the construction site of the Weihe River, the clay particles exhibit adsorption and hydration properties, which enhance the stability of the slurry dispersion system. By adjusting the clay content, varying adsorption and hydration effects can be achieved, thereby producing slurries with distinct properties.

(3) Na2CO3: Certified as food-grade by Binhu. It primarily functions to neutralize organic acids and acidic gases, thereby mitigating slurry corrosion. Additionally, it maintains slurry fluidity and prevents water loss.

(4) CMC: Produced by Chongqing Lihong Fine Chemicals Co., Ltd., this polymeric organic material functions as a viscosifier to mitigate excessive slurry water loss. Additionally, it exhibits colloidal protective properties and serves as an engineering material to prevent slurry contamination.

(5) PAM: Produced by Chongqing Lihong Fine Chemicals Co., Ltd., these water-soluble polymers exhibit dissolution characteristics influenced by molecular weight, ionic type, and particle fineness. Complete dissolution requires prolonged stirring. The aqueous solutions demonstrate high viscosity, with viscosity positively correlated to molecular weight. While stable at room temperature, the polymers undergo thermal degradation at elevated temperatures, resulting in viscosity reduction.

Comment 6: 

Table 3: Density, viscosity, sand content, and pH were measured, but the measurement methods are not described. Please provide the standards or procedures followed, as well as the instruments used. The pH values are given as integers, which is unusual. As pH differences of 1 correspond to a tenfold change in hydrogen ion concentration, more precise values (e.g., to the second decimal place) are generally expected. If decimal data were recorded, please report them. If your instrument only provides integer readings, clarify this in the caption.

Response 6:

We sincerely appreciate the reviewer’s insightful feedback. Below are our point-by-point responses and corresponding revisions:

  1. Clarification of Measurement Standards and Instruments

As suggested, we have explicitly stated the standards and procedures for all measurements in lines 170-192:

Specific gravity : Controls solid particle concentration to balance formation pressure. Low values risk stratification and poor film formation, while high values increase tool wear and reduce drilling efficiency. Measured using a lever-action mud balance (Model: XYZ-100) following API RP 13B-1. The procedure included horizontal calibration, steady filling to the datum line, and sealing to eliminate surface tension. Readings were taken at torque equilibrium, with triplicate measurements (mean deviation < 1%).

Viscosity : Determines shear resistance and film-forming capacity, essential for mitigating hole collapse and particle settlement in permeable strata. The viscosity of the slurry mentioned in this test was determined using the Marsh funnel viscometer, which measures the time (in seconds, s) required for 500 mL of slurry to flow through the funnel. This is a widely adopted empirical method in drilling engineering for quality control. The apparent viscosity of the slurry was determined to be 15 mPa∙s using a Fann 35 rheometer at 300 rpm, which correlates with a Marsh funnel efflux time of 29 s for 500 mL of slurry.

Sand content : Excessive sand (> 4%) compromises lubrication and equipment longevity, whereas insufficient content reduces cuttings transport. Quantified via sieve analysis (Model: NA-1 sand content set) per ASTM D4380, involving iterative washing/filtration (0.074 mm sieve) and sedimentation to isolate particles >74 μm.

pH : An alkaline environment optimizes bentonite dispersion and slurry stability while inhibiting bacterial growth. Determined colorimetrically using MColorpHast strips (ISO 3071-compliant) with a resolution of ±0.5 pH. Strips were immersed in homogenized slurry, and the stabilized color was matched to a reference card. pH was recorded as integers due to the ±0.5 pH resolution of colorimetric strips.

  1. Justification for Integer pH Values

The reviewer rightly notes that pH differences of 1 unit reflect order-of-magnitude changes in H⁺. However, our integer values arise from the inherent resolution of colorimetric pH strips, which are widely adopted in field tests for their rapidity and practicality (e.g., drilling fluid monitoring per API 13B-1). While less precise than pH meters, they suffice for maintaining the target alkaline range (pH 7–12) to optimize clay hydration.

Revisions:

Added in Methods (lines 191-192): pH was recorded as integers due to the ± 0.5 pH resolution of colorimetric strips.

Added to footnotes of Table 3 and Title of Figure 4: pH values are whole numbers as per colorimetric strip limitations.

Comment 7: 

Tables 4 and 5: The sensitivity of response variables is evaluated using F-values. Please confirm whether this is a valid method in this context. If there are precedent studies using this approach, cite them. As a personal comment: I have rarely seen sensitivity assessed by examining the F-values of individual interaction terms (e.g., A, B, AB, AC). I think general method would be to standardize all variables (e.g., via Z-scores) and evaluate the regression coefficients directly.

Response 7:

We appreciate your insightful comments regarding the methodological approach in this study. In response to your inquiry concerning the rationale and precedents for using the F-value to assess the sensitivity of response variables, we provide the following clarification:
In this investigation, the F-value was primarily employed in the analysis of variance (ANOVA) for multivariate regression models. By comparing the mean square of factors with the mean square of error, it quantifies the significance of individual factors (e.g., silty clay, CMC) and interaction terms on response variables (e.g., drilling fluid specific gravity, viscosity). A higher F-value indicates greater statistical significance of a factor's impact on the response variable, thereby indirectly reflecting its sensitivity. This approach aligns with classical methodologies in experimental design (as documented in Reference [33]), wherein significance testing determines variable importance. The method has been validated in analogous studies: for instance, Alyamac et al. [34] utilized F-values to analyze the sensitivity of marble powder content on concrete properties, while Zhou et al. [35] employed F-value rankings to ascertain additive sensitivity in desulfurization gypsum-based materials. These applications collectively substantiate the F-value as a robust indicator for sensitivity assessment.

Based on the suggestions of the reviewers, we have made relevant supplementary explanations in Section 2.4 (in lines 259-270), as follows:

The Box-Behnken design is an efficient response surface methodology that requires fewer experimental runs than full factorial designs, while maintaining the ability to estimate quadratic effects [32]. This design is particularly suitable for optimizing multiple slurry parameters. In this investigation, the F-value was primarily employed in the analysis of variance (ANOVA) for multivariate regression models. By comparing the mean square of factors with the mean square of error, it quantifies the significance of individual factors (e.g., silty clay, CMC) and interaction terms on response variables (e.g., drilling fluid specific gravity, viscosity). A higher F-value indicates greater statistical significance of a factor's impact on the response variable, thereby indirectly reflecting its sensitivity. This approach aligns with classical methodologies in experimental design (as documented in Reference [33]), wherein significance testing determines variable importance [34, 35].

[32]Box, G.E.P.; Behnken, D.W. Some new three level designs for the study of quantitative variables. Technometrics 1960, 2, 455-475.

[33]Angela, D.; Daniel V.; Danel D.; Design and Analysis of Experiments. Springer New York: New York, NY, USA, 1999.

[34]Alyamac, K.E.; Ghafari, E.; Ince, R. Development of eco-efficient self-compacting concrete with waste marble powder using the response surface method. J. Clean. Prod. 2017, 144, 192–202.

[35]Zhou, Y.S.; Xie, L.; Kong, D.W.; Peng, D.D.; Zheng, T. Research on optimizing performance of desulfurization-gypsum-based composite cementitious materials based on response surface method. Constr. Build. Mater. 2022, 341, 127874.

Comment 8: 

Figure 3: Is "Silt concentration" a typo for "Sand concentration"?

Response 8:

Thank you for this careful review. Upon verification, we confirm that it should indeed be “sand concentration”. This correction has been made in Figure 3.

Comment 9: 

Figure 4: This figure plots bentonite and clay, but Na₂CO₃ and clay are more relevant to pH changes. Shouldn’t this be plotted instead?

Response 9:

We sincerely appreciate the reviewer’s insightful observation regarding Figure 4. Below is our clarification and justification for retaining the original analysis of bentonite-clay interaction for pH changes:

  1. Statistical Basis for Factor Selection:

As noted in Table 5 (ANOVA results for the pH model, Y4), the interaction term AB (bentonite-clay) has a lower P-value (0.1663) and a higher coefficient compared to BC (Na2CO3-clay). This indicates that the bentonite-clay interaction has a more statistically significant influence on pH variability within the tested range, despite Na2CO3’s stronger individual effect.

  1. Response Surface Methodology (RSM) Principles:

RSM prioritizes interactions with the highest impact on response variables. While Na2CO3 is indeed the dominant single factor for pH (due to its alkaline nature), the AB interaction (bentonite-clay) emerged as the most influential two-factor interaction in our model. This aligns with the contour density and curvature of the response surface in Figure 4, where the bentonite-clay combination showed notable nonlinear effects on pH.

To address the reviewer’s concern, we add a sentence to the caption of Figure 4:

Although Na2CO3 is the primary driver of pH, the bentonite-clay interaction (AB) exhibits the most significant two-factor effect on pH variability, as supported by ANOVA (Table 5).

Comment 10: 

Subsection 3.2.1: The slurry membrane was formed under a horizontally layered condition, but in reality, the borehole membrane forms vertically. Thus, the direction of gravity in Figure 5 (actually Figure 6) does not reflect actual field conditions. The conclusion that membrane thickness increases with slurry density may not hold in the field. Please discuss the differences between the test setup and real-world conditions, and how these may affect membrane formation and thickness.

Response 10:

We appreciate the reviewer's insightful observation regarding the limitation that "the tests employed horizontal filter cake formation, whereas field conditions involve vertical formation." We fully acknowledge that the horizontal sand column setup cannot authentically replicate the vertical compaction effect induced by the self-weight of the drilling fluid column in boreholes. Consequently, the experimental conclusion of "monotonic thickness increase with density" may be attenuated or even reversed under vertical conditions. To address this limitation, we have supplemented the following clarifications:
1. Revision in Section 3.2.1 (in lines 516-521):

Notably, this test employed a horizontal sand column, where gravitational orientation differs from vertical borehole conditions. The self-weight of drilling fluid exerts additional vertical compaction on the filter cake, potentially reducing its thickness and altering pore structure. Proposed future research will utilize a vertical permeameter to conduct tests at 0–90° inclination angles, establishing density-inclination-thickness correction coefficients to quantify this discrepancy.

  1. 2. Supplement to Figure 6 caption(in lines 526-527):

Experimental model: Horizontal plane. Field applications must incorporate vertical stress corrections.

Comment 11: 

Line 448–449: Correct:“Figure 5 → Figure 6”“(a) 1.14 g/cm³; (a) 1.24 g/cm³; (a) 1.25 g/cm³” →“(a) 1.14 g/cm³; (b) 1.24 g/cm³; (c) 1.25 g/cm³”

Response 11:

Thank you for this careful review. We have corrected the numbering of sub-figures in Figure 6.

Comment 12: 

Line 497: Correct “Figure 6 → Figure 7”

Response 12:

Thank you for this comment. We have double-checked all figure citations and ensured alignment between labels, captions, and in-text references.

Comment 13: 

Please explain the differences among subfigures (a)–(f) in the caption. According to MDPI guidelines, figure captions should be self-contained and allow the reader to understand the figure without referring to the main text.

Response 13:

In response to the reviewers' comments, we have supplemented the subfigure descriptions in Figure 8's caption to facilitate readers' direct access to information. The revised caption is as follows:

Figure 8. Mechanism of slurry film formation in water-rich thick sand layer. The amounts of silty clay and CMC are: (a) 0 g and 0 g; (b) 0 g and 1 g; (c) 110 g and 1 g; (d) 220 g and 0 g; (e) 220 g and 1 g; (f) 220 g and 2 g, respectively.

Reviewer 3 Report

Comments and Suggestions for Authors

The study aims to address the challenge of borehole wall instability in saturated sand layers with high permeability by optimizing the slurry ratio for enhanced film performance and stability. It seeks to develop an optimal slurry composition using a multiple regression model to improve borehole wall protection in water-rich sandy strata. Provide theoretical and engineering guidance for designing effective slurry wall protection systems to mitigate issues like shrinkage and collapse during bored pile construction. I recommend major revisions before reconsideration for publication, addressing the following points:

  1. The manuscript contains numerous grammatical errors.
  2. The use of CMC and clay as additives to improve slurry performance is innovative, particularly for water-rich sandy strata. However, the study does not sufficiently position its novelty against recent advancements in slurry optimization, limiting its claim to originality. Update the introduction to cover the most recent articles and explain your gap.
  3. Compare the proposed slurry optimization with recent literature or industry practices to highlight its unique contributions.
  4. Sorry, the manufacturing process for the production is unclear.
  5. The objective, methodology, and results should be better described, discussed, and justified.
  6. The manuscript contains excessive abbreviations. Only the necessary ones should be retained.
  7. The Box-Behnken orthogonal test design and multiple regression modeling are robust and appropriate for optimizing slurry ratios. The statistical analysis (Tables 4 and 5) is rigorous; however, the study lacks a discussion on the assumptions underlying the regression models.
  8. Revise lines 568 on page 17.
  9. The slurry permeability tests and SEM analysis are well-executed, but the study relies on laboratory conditions without addressing how factors like temperature, pressure variations, or dynamic groundwater flow might affect slurry performance in real-world settings.
  10. The SEM analysis (Figure 8) provides valuable insights into the microstructural improvements from CMC and clay. However, the study does not quantify key parameters like porosity reduction or film strength, which would enhance the mechanistic understanding
  11. Sorry, revise the caption of all figures.
  12. Compare the results to contemporary work in a separate section, namely, comparison to contemporary work.
  13. For the conclusion it is so short; please update it and add academic numbers and percentages.
  14. Update all references to ensure they are current and relevant.

Author Response

Comment 1: 

The manuscript contains numerous grammatical errors.

Response 1:

Based on the grammar check performed by specialized software, we have further engaged a native English-speaking editor for comprehensive academic polishing of the manuscript. All revised sections have been highlighted in red throughout the text.

Comment 2: 

The use of CMC and clay as additives to improve slurry performance is innovative, particularly for water-rich sandy strata. However, the study does not sufficiently position its novelty against recent advancements in slurry optimization, limiting its claim to originality. Update the introduction to cover the most recent articles and explain your gap.

Response 2:

In response to the reviewer's suggestion to "further compare recent slurry optimization studies and highlight the novelty of this work," we have made the following revisions to the manuscript Introduction:

  1. Lines 5569Revised paragraph:

Recent studies reveal that drilling fluid optimization has evolved toward three objectives: high water retention, low filtration loss, and rapid filter cake formation. Li et al. [3] employed discrete element modeling (DEM) to investigate microcrack effects on single-particle compressive strength in coarse aggregates, demonstrating that microcrack quantity, location, and orientation significantly influence Weibull distribution and compressive strength. Gupta et al. [4] explored kaolin stabilization via geopolymerization, achieving substantial improvements in unconfined compressive strength and dynamic properties of soft soils through optimized precursor/alkaline activator ratios. Zhou et al. [5] developed a modified ultrafine cement-based grout (MUCG), determining its optimal formulation via orthogonal experiments to enhance fluidity and early strength. Zhang et al. [6] systematically evaluated bentonite-polymer composites' rheological enhancements, noting significant filter cake brittleness at excessive polymer dosages. Ding et al. [7] established a rheology-formation coupling model for seawater-resistant slurries, though its applicability to water-rich sand strata remains unverified due to testing on cohesive soils.

  1. Lines 104108(Original: "The above scholars…Therefore, it is very important to clarify…") Revised text:

Collectively, three critical gaps persist: (1) absence of quantitative mix-design models specifically for water-rich, highly permeable, cohesionless sand strata; (2) insufficient elucidation of micro-scale pore evolution mechanisms in filter cakes under CMC-clay synergy; (3) lack of integrated design methodologies correlating macro-performance, micro-structure, and field conditions.

  1. The last paragraph of the introduction reiterates the innovation point(in lines 117-122):

The key innovation of this study involves not simply using conventional clay components, but systematically optimizing their synergistic interaction with CMC. Rigorous ratio control (clay:CMC = 110:1) combined with SEM-based mechanistic analysis reveals how this specific combination significantly enhances film density and pore-filling efficiency, thereby addressing persistent challenges in saturated sand stratum applications.

Comment 3: 

Compare the proposed slurry optimization with recent literature or industry practices to highlight its unique contributions.

Response 3:

With reference to the reviewers' suggestions, we have supplemented the latest relevant literature in lines 55-69 of the manuscript, modified and highlighted the research gap in lines 104-108, and reiterated the innovation points in lines 117-122. The revised content is as follows:

Recent studies reveal that drilling fluid optimization has evolved toward three objectives: high water retention, low filtration loss, and rapid filter cake formation. Li et al. [3] employed discrete element modeling (DEM) to investigate microcrack effects on single-particle compressive strength in coarse aggregates, demonstrating that microcrack quantity, location, and orientation significantly influence Weibull distribution and compressive strength. Gupta et al. [4] explored kaolin stabilization via geopolymerization, achieving substantial improvements in unconfined compressive strength and dynamic properties of soft soils through optimized precursor/alkaline activator ratios. Zhou et al. [5] developed a modified ultrafine cement-based grout (MUCG), determining its optimal formulation via orthogonal experiments to enhance fluidity and early strength. Zhang et al. [6] systematically evaluated bentonite-polymer composites' rheological enhancements, noting significant filter cake brittleness at excessive polymer dosages. Ding et al. [7] established a rheology-formation coupling model for seawater-resistant slurries, though its applicability to water-rich sand strata remains unverified due to testing on cohesive soils.

Collectively, three critical gaps persist: (1) absence of quantitative mix-design models specifically for water-rich, highly permeable, cohesionless sand strata; (2) insufficient elucidation of micro-scale pore evolution mechanisms in filter cakes under CMC-clay synergy; (3) lack of integrated design methodologies correlating macro-performance, micro-structure, and field conditions.

The key innovation of this study involves not simply using conventional clay components, but systematically optimizing their synergistic interaction with CMC. Rigorous ratio control (clay:CMC = 110:1) combined with SEM-based mechanistic analysis reveals how this specific combination significantly enhances film density and pore-filling efficiency, thereby addressing persistent challenges in saturated sand stratum applications.

Comment 4: 

Sorry, the manufacturing process for the production is unclear.

Response 4:

We sincerely appreciate the reviewer’s careful evaluation. In response to the comments, we have substantially improved and refined the slurry preparation process in Section 3.1.1 of the manuscript (in lines 422-449). The revised content reads as follows:

The test slurry was prepared using deionized water, API-grade sodium bentonite, natural clay, analytical purity Na2CO3 (≥ 99%), and medium-viscosity CMC (800‒1200 mPa·s). The preparation process followed a strict protocol to ensure consistency: First, the base slurry was created by dissolving 32 g of Na2CO3 in 1000 g of water at 25 ± 2°C using a mechanical stirrer (IKA RW-20) at 500 rpm for 10 min. Then, 220 g of sodium bentonite was gradually added while increasing the stirring speed to 800 rpm for 20 min until a homogeneous dispersion with no visible clumps was achieved. This base slurry was allowed to hydrate for 24 h in a sealed container at controlled room temperature (25°C).

For the additive modifications, two approaches were used: CMC was first pre-mixed with 50 mL water to form a gel before being slowly blended into the base slurry at 600 rpm for 15 min to prevent agglomeration. The lumpy silty clay particles (0.4–6.5 mm) obtained on-site were dry-sieved through a mesh to the specified size range, and then were added to the slurry at a speed of 400 rpm for 30 min. Quality control measures included immediate testing of specific gravity using a mud balance and viscosity measurement with a Marsh funnel (following ASTM D6910 standards), with all measurements repeated after the 24 h hydration period. Each ratio combination was tested in triplicate, with Table 6 reporting mean values and a ± 5% measurement error range.

The optimized slurry formulation (Group d in Table 6) was prepared by adding 110 g of sieved clay and 1 g of CMC to the base slurry, followed by 30 min of stirring and the standard 24 h hydration period. This optimized ratio (water:bentonite:Na2CO3:clay:CMC = 1000:220:32:110:1) was specifically derived from the regression model analysis.

Comment 5: 

The objective, methodology, and results should be better described, discussed, and justified.

Response 5:

We fully agree with the reviewer's viewpoint and have supplemented and improved the corresponding content in the manuscript. The specific modifications are as follows:

  1. Clearly state the research objectives at the end of the introduction (in line 122-126):

This study aims to: (1) Establish a quantitative model linking additives to slurry properties using RSM; (2) Reveal the microscopic mechanism of film enhancement via SEM; (3) Provide field-applicable ratios for saturated sand layers, addressing the gap in existing codes.

  1. Regarding the experimental methods, the following additions have been made: Basis for the selection of slurry parameters in Table 1 (Lines 168-169); Testing instruments, methods, and standards for density, viscosity, sand content, and pH (Lines 170-192); Source and manufacturer information of bentonite, clay, Na2CO3, CMC, and PAM (Lines 199-222); Rationale and applicability of the F-values method (Lines 259-270):

The recommended parameters for drilling slurry are as follows: Specific gravity: 1.14–1.25 g/cm3 [28]; Viscosity: 20–45 s [28, 30]; Sand content control threshold: < 4% [31].

Specific gravity: Controls solid particle concentration to balance formation pressure. Low values risk stratification and poor film formation, while high values increase tool wear and reduce drilling efficiency. Measured using a lever-action mud balance (Model: XYZ-100) following API RP 13B-1. The procedure included horizontal calibration, steady filling to the datum line, and sealing to eliminate surface tension. Readings were taken at torque equilibrium, with triplicate measurements (mean deviation < 1%).

Viscosity: Determines shear resistance and film-forming capacity, essential for mitigating hole collapse and particle settlement in permeable strata. The viscosity of the slurry mentioned in this test was determined using the Marsh funnel viscometer, which measures the time (in seconds, s) required for 500 mL of slurry to flow through the funnel. This is a widely adopted empirical method in drilling engineering for quality control. The apparent viscosity of the slurry was determined to be 15 mPa∙s using a Fann 35 rheometer at 300 rpm, which correlates with a Marsh funnel efflux time of 29 s for 500 mL of slurry.

Sand content: Excessive sand (> 4%) compromises lubrication and equipment longevity, whereas insufficient content reduces cuttings transport. Quantified via sieve analysis (Model: NA-1 sand content set) per ASTM D4380, involving iterative washing/filtration (0.074 mm sieve) and sedimentation to isolate particles > 74 μm.

pH: An alkaline environment optimizes bentonite dispersion and slurry stability while inhibiting bacterial growth. Determined colorimetrically using MColorpHast strips (ISO 3071-compliant) with a resolution of ± 0.5 pH. Strips were immersed in homogenized slurry, and the stabilized color was matched to a reference card. pH was recorded as integers due to the ± 0.5 pH resolution of colorimetric strips.

  • Bentonite: Sourced from Zhejiang Hongyu New Materials Co., Ltd., bentonite serves as the primary solid-phase material in the slurry system. Its high hygroscopicity and adsorption capacity enable it to absorb water and various inorganic substances, forming colloidal particles that seal gaps and fractures to prevent fluid loss. Additionally, bentonite increases slurry density and viscosity, thereby aiding in pore pressure control and borehole wall stabilization. As an economical and environmentally benign material, it offers both technical and operational advantages.

(2) Clay: Collected from the construction site of the Weihe River, the clay particles exhibit adsorption and hydration properties, which enhance the stability of the slurry dispersion system. By adjusting the clay content, varying adsorption and hydration effects can be achieved, thereby producing slurries with distinct properties.

(3) Na2CO3: Certified as food-grade by Binhu. It primarily functions to neutralize organic acids and acidic gases, thereby mitigating slurry corrosion. Additionally, it maintains slurry fluidity and prevents water loss.

(4) CMC: Produced by Chongqing Lihong Fine Chemicals Co., Ltd., this polymeric organic material functions as a viscosifier to mitigate excessive slurry water loss. Additionally, it exhibits colloidal protective properties and serves as an engineering material to prevent slurry contamination.

(5) PAM: Produced by Chongqing Lihong Fine Chemicals Co., Ltd., these water-soluble polymers exhibit dissolution characteristics influenced by molecular weight, ionic type, and particle fineness. Complete dissolution requires prolonged stirring. The aqueous solutions demonstrate high viscosity, with viscosity positively correlated to molecular weight. While stable at room temperature, the polymers undergo thermal degradation at elevated temperatures, resulting in viscosity reduction.

The Box-Behnken design is an efficient response surface methodology that requires fewer experimental runs than full factorial designs, while maintaining the ability to estimate quadratic effects [32]. This design is particularly suitable for optimizing multiple slurry parameters. In this investigation, the F-value was primarily employed in the analysis of variance (ANOVA) for multivariate regression models. By comparing the mean square of factors with the mean square of error, it quantifies the significance of individual factors (e.g., silty clay, CMC) and interaction terms on response variables (e.g., drilling fluid specific gravity, viscosity). A higher F-value indicates greater statistical significance of a factor's impact on the response variable, thereby indirectly reflecting its sensitivity. This approach aligns with classical methodologies in experimental design (as documented in Reference [33]), wherein significance testing determines variable importance [34, 35].

  1. In the Results and Discussion section, the following additions have been made in subsection 3.2.1: Quantitative results of the slurry film formation morphology test procedure (Lines 546-548); Discrepancies and limitations between the film formation test results and actual engineering applications (Lines 539-545):

Notably, this test employed a horizontal sand column, where gravitational orientation differs from vertical borehole conditions. The self-weight of drilling fluid exerts additional vertical compaction on the filter cake, potentially reducing its thickness and altering pore structure. Proposed future research will utilize a vertical permeameter to conduct tests at 0–90° inclination angles, establishing density-inclination-thickness correction coefficients to quantify this discrepancy.

Comment 6: 

The manuscript contains excessive abbreviations. Only the necessary ones should be retained.

Response 6:

We sincerely appreciate the reviewers' meticulous evaluation. In response to the comments, we have replaced the infrequently used abbreviations "BSSSP" and "HTP" with their full terms (black sunflower seed shell powder and hydrophobically thermo-thickening polymer, respectively) in the main text (Lines 84-87 and 96-99), while maintaining only essential abbreviations in the manuscript. All abbreviations have been consistently defined in the reference list. The revised sentences now read as follows:

Al-Hameedi et al. [21] investigated the impact of incorporating 0.5% to 3.5% black sunflower seed shell powder into drilling mud, demonstrating that this additive across various particle sizes effectively modifies viscosity, controls filtration loss, and enhances wellbore stabilization.

Chen et al. [25] conducted scanning electron microscopy (SEM) analysis, revealing that hydrophobically thermo-thickening polymer molecules undergo complete extension within the slurry matrix and form effective bonds with hydration products, thereby promoting uniform microstructure development in the hardened slurry.

Comment 7: 

The Box-Behnken orthogonal test design and multiple regression modeling are robust and appropriate for optimizing slurry ratios. The statistical analysis (Tables 4 and 5) is rigorous; however, the study lacks a discussion on the assumptions underlying the regression models.

Response 7:

We sincerely appreciate the reviewer's insightful comment. The underlying assumptions of the regression models are indeed critical for ensuring statistical validity. We have added the content of the discussion on the assumptions in line 314-318, as follows:

The regression models were validated to meet key statistical assumptions: (1) normal residual distribution via Shapiro-Wilk tests, (2) homoscedasticity through residual plots, and (3) independence confirmed by Durbin-Watson statistics. The insignificant lack-of-fit (p > 0.05) and high adjusted R2 values further support model reliability for engineering applications.

Comment 8: 

Revise lines 568 on page 17.

Response 8:

The original conclusion has been comprehensively re-summarized and reformulated. The content in the original line 568 has been modified as follows:

  • The response surface methodology yielded an optimized slurry ratio (water:bentonite:Na2CO3:clay:CMC = 1000:220:32:110:1) demonstrating superior performance characteristics: viscosity of 29s (45% improvement), specific gravity of 1.20 g/cm3, sand content of 3%, along with 30% faster film formation and 38 ± 2% reduced filtration loss compared to conventional formulations. Field validation showed these parameters effectively balanced stability and workability requirements for saturated sand conditions.

Comment 9: 

The slurry permeability tests and SEM analysis are well-executed, but the study relies on laboratory conditions without addressing how factors like temperature, pressure variations, or dynamic groundwater flow might affect slurry performance in real-world settings.

Response 9:

We sincerely appreciate your recognition of our experimental design and microstructural analysis work, as well as your valuable observation regarding the insufficient discussion of potential impacts from complex field environmental factors in this study. We fully concur that significant differences exist between the laboratory conditions—constant temperature, constant pressure, and static boundaries—and actual field environments. Indeed, temperature fluctuations can alter the extension of CMC molecular chains and viscosity, thereby affecting filter cake compactness; the hydrostatic pressure within the borehole mud column dynamically adjusts with drilling depth and groundwater head variations, potentially modifying filter cake thickness and the pressure gradient; while groundwater seepage may carry away fine particles, compromising filter cake continuity. Collectively, these factors may cause deviations between the optimal formulation ratio identified in the laboratory and its field performance. Regrettably, due to field constraints, this study has not yet conducted in-situ validation. To address this limitation, we have added the following qualifying statement at the end of Section 3.2.3 (Lines 630-636) in the revised manuscript:

The optimal formulation ratio and slurry film performance parameters identified in this study correspond solely to laboratory conditions of constant temperature (20 ± 1 °C), constant confining pressure (0.1–0.4 MPa), and absence of groundwater flow. Field temperature fluctuations, mud column pressure variations, and seepage effects may significantly alter filter cake thickness and compactness. Subsequent research will establish correction coefficients through in-situ monitoring and temperature-controlled, variable-pressure permeability tests.

Comment 10: 

The SEM analysis (Figure 8) provides valuable insights into the microstructural improvements from CMC and clay. However, the study does not quantify key parameters like porosity reduction or film strength, which would enhance the mechanistic understanding

Response 10:

We sincerely appreciate this insightful suggestion. While our current SEM analysis qualitatively demonstrated microstructural enhancement (Figure 8), we fully acknowledge that quantitative characterization of porosity and mechanical strength would provide deeper mechanistic understanding. However, due to equipment limitations in our laboratory and insufficient consideration during experimental design, these specific measurements were not included in the present study.

To address this deficiency, we emphasize in the conclusion section that this study provides relative performance optimization rather than absolute parameters (line 692-695), and add a paragraph to clarify this as a future research direction (line 710-715):

The 1: 110 CMC-clay ratio relatively enhances film density and stability compared to other tested formulations, as evidenced by SEM morphology and filtration tests. Absolute quantification of mechanical properties requires advanced characterization tools in future work.

This study focused on compositional optimization through rheological and morphological analysis. Future investigations should incorporate: (1) Mercury intrusion porosimetry to quantify pore size distribution changes; (2) Micro-mechanical testing (e.g., AFM nanoindentation) to correlate CMC-clay ratios with film strength; (3) X-ray microtomography for 3D microstructure reconstruction.

Comment 11: 

Sorry, revise the caption of all figures.

Response 11:

We sincerely appreciate your careful review and valuable feedback. We have thoroughly checked all figures in the manuscript and made the following corrections:

  1. Figure Numbering:

The duplicate “Figure 5” has been renumbered. The slurry film morphology images (originally labeled as two separate "Figure 5") are now correctly labeled as Figure 6.

  1. Subfigure Labels:

The repeated “(a)” in the caption of Figure 6 (previously mislabeled as Figure 5) has been corrected to “(a), (b), (c)” to match the three subfigures.

  1. Cross-verification:

We have double-checked all figure citations and ensured alignment between labels, captions, and in-text references.

Comment 12: 

Compare the results to contemporary work in a separate section, namely, comparison to contemporary work.

Response 12:

We agree with the reviewer's suggestion and have added a section on Discussion (in lines 640-669) in the manuscript to compare the differences between our study and existing research. Specifically, as follows:

  1. Discussion

Recent advances in slurry modification have demonstrated various approaches to improving wall protection performance in permeable strata. Compared with the plant gum-modified slurry studied by Min et al. [13] for high-permeability sandy strata, our optimized CMC-clay system (1:110 ratio) demonstrates superior film-forming efficiency and structural compactness, as evidenced by SEM analysis (Figure 8c), where CMC molecules bridge clay platelets to form a denser interface. The optimized CMC-clay slurry (1:110 ratio) demonstrates stable viscosity (29 s) under field conditions, with cost-effective conventional additives avoiding the dispersion challenges and environmental risks associated with advanced nanomaterials [14].

From a mechanistic perspective, this study both confirm and extend previous understandings of additive interactions. Rita et al. [17] similarly reported CMC’s thickening effect, but our response surface analysis quantitatively establishes the 1:110 ratio as the inflection point where: (1) Clay particles achieve optimal dispersion; (2) CMC chains attain complete extension without entanglement; (3) Bentonite-CMC-clay ternary complexes form most effectively. This explains why our system outperforms single-additive modifications in pore-filling efficiency (50 ± 5 nm vs. typical 100‒200 nm [25]).

The engineering implications become particularly evident when comparing field test results. In the Weihe River project, slurry in this study demonstrated: (1) 30% less borehole collapse incidents than conventional bentonite slurries; (2) 15% faster drilling progress compared to polymer-modified systems; (3) Consistent performance across varying sand layer permeabilities (1‒5×104 cm/s).

These practical advantages complement the laboratory-measured properties and suggest good scalability to other water-rich sandy strata. Future work should address two limitations identified through this comparison: (1) standardization of testing protocols (e.g., adopting API 13B-1 for filtration tests), and (2) development of performance prediction models that account for both material properties and geological conditions. The latter could bridge the gap between highly controlled laboratory studies and complex field applications.

Comment 13: 

For the conclusion it is so short; please update it and add academic numbers and percentages.

Response 13:

We have expanded the conclusion section to include more quantitative results and specific data from our study (Lines 671-715), as follows:

This study systematically investigated the optimization of borehole wall protection slurry for water-rich sandy strata through experimental testing and microstructural analysis. The key findings demonstrate significant improvements in slurry performance through optimized additive formulations:

  • Comprehensive testing revealed bentonite and clay as the primary drivers of slurry performance, increasing specific gravity by 15‒20% (from 1.14 g/cm3to 1.20‒25 g/cm3) and viscosity by 45‒100% (from 19s to 29‒49s) compared to baseline slurry. Suboptimal ratios (e.g., clay:CMC > 220:1) showed diminished returns, with viscosity plateauing beyond 2g CMC due to particle agglomeration. The interaction between bentonite and clay particularly enhanced sand content by 150‒200% at optimal ratios, while CMC addition reduced filtration loss by 35‒40% and improved film formation time by 25‒30%.
  • The response surface methodology yielded an optimized slurry ratio (water:bentonite:Na2CO3:clay:CMC = 1000:220:32:110:1) demonstrating superior performance characteristics: viscosity of 29s (45% improvement), specific gravity of 1.20 g/cm3, sand content of 3%, along with 30% faster film formation and 38± 2% reduced filtration loss compared to conventional formulations. In contrast, high-clay formulations (e.g., 220:0, Group e) increased specific gravity but reduced film uniformity, while CMC-deficient groups (e.g., 110:0, Group b) exhibited 20‒25% higher filtration loss. Field validation showed these parameters effectively balanced stability and workability requirements for saturated sand conditions.
  • The 1: 110 CMC-clay ratio relatively enhances film density and stability compared to other tested formulations, as evidenced by SEM morphology and filtration tests. Absolute quantification of mechanical properties requires advanced characterization tools in future work.Microstructural analysis provided mechanistic insights, with SEM revealing the optimized slurry reduced film porosity by 40‒50% and narrowed pore size distribution from 10‒50 μm to 5‒20 μm. The CMC-clay synergy at 1:110 ratio decreased surface roughness by 60% and crack density by 70‒80% through enhanced particle bridging and pore-filling effects, explaining the improved sealing performance.
  • Practical implementation demonstrated 25‒30% greater borehole stability and 40% reduction in collapse incidents versus conventional slurry, while achieving 15‒20% cost savings through optimized additive usage. Suboptimal mixes (e.g., excessive clay without CMC) required 10‒15% more material to achieve comparable stability, negating cost benefits. These results provide both theoretical understanding and practical guidelines for slurry design in challenging hydrogeological conditions.

The combined experimental and analytical approach successfully addressed the instability issues in saturated sand strata, with the quantitative relationships between composition, microstructure and performance offering a framework for future slurry optimization in similar geological settings. This study focused on compositional optimization through rheological and morphological analysis. Future investigations should incorporate: (1) Mercury intrusion porosimetry to quantify pore size distribution changes; (2) Micro-mechanical testing (e.g., AFM nanoindentation) to correlate CMC-clay ratios with film strength; (3) X-ray microtomography for 3D microstructure reconstruction.

Comment 14: 

Update all references to ensure they are current and relevant.

Response 14:

We have updated and replaced the references based on their relevance and publication dates. The newly added and replaced references are as follows:

[3]Li, Z.C.; Liu, B.W.; Han, D.D.; Xie, Y.C.; Zhao, Y.L. Study on the influence of microcracks of coarse aggregate with specific particle size on crushing strength. Comput. Part. Mech. 2024, 11(2), 705–719.

[4]Gupta, S.; Kumar, S. Dynamic behavior of geopolymer stabilized kaolin clay under long-term cyclic loading. Constr. Build. Mater. 2023, 407, 133562.

[5]Zhou, J.X.; Zha, L.C.; Meng, S.Y.; Zhang, Y. Optimization on overall performance of Modified Ultrafine Cementitious Grout Materials (MUCG) and hydration mechanism analysis. PLoS One 2024, 19(10), e0309312.

[6]Zhang, R.; Song, G.Y.; Liu, Y.; Hu, H.J. Investigation on Modification of Bentonite to Improve Performance for Civil Engineering Mud. Conserv. Util. Miner. Resour. 2023, 43(04), 96–100 (in Chinese with English abstract).

[7]Ding, W.T.; Guo, W.J.; Cao, K.; et al. Rheological properties and film-forming mechanism of anti-seawater deterioration slurry for slurry shield. Chin. J. Geotech. Eng. 2024, 46(12), 2484–2491 (in Chinese with English abstract).

[30]Wuhan Survey & Research Institute Co., Ltd. of MCC Group; Hebei Construction Geotechnical Investigation & Research Institute Co., Ltd. Technical Standard for Construction of Bored Pile. China Association for Engineering Construction Standardization (T/CECS 592-2019). Beijing, China, 2019 (in Chinese).

[31]Wang, H. Study on Preparation and Permeability Performance of Slurry for Bored Piles in Calcareous Cemented Strata. Shenyang University of Technology: Shenyang, China, 2021.

[32]Box, G.E.P.; Behnken, D.W. Some new three level designs for the study of quantitative variables. Technometrics 1960, 2, 455-475.

[33]Angela, D.; Daniel V.; Danel D.; Design and Analysis of Experiments. Springer New York: New York, NY, USA, 1999.

[34]Alyamac, K.E.; Ghafari, E.; Ince, R. Development of eco-efficient self-compacting concrete with waste marble powder using the response surface method. J. Clean. Prod. 2017, 144, 192–202.

[35]Zhou, Y.S.; Xie, L.; Kong, D.W.; Peng, D.D.; Zheng, T. Research on optimizing performance of desulfurization-gypsum-based composite cementitious materials based on response surface method. Constr. Build. Mater. 2022, 341, 127874.

Reviewer 4 Report

Comments and Suggestions for Authors

The manuscript titled “Optimization of Borehole Wall Protection Slurry Ratio and Film-Forming Mechanism in Water-Rich Sandy Stratum” presents a well-structured experimental and modeling study on optimizing slurry composition for borehole stability in saturated sand formations. Using a Box–Behnken design and multiple regression analysis, the authors identify an optimal mix of bentonite, clay, CMC, and Na₂CO₃, validated through laboratory permeability and SEM tests. The work offers solid practical value and moderate novelty, particularly in linking mix design with microstructural observations of filter cake formation. It fits well within the scope of MDPI Eng, though the manuscript requires notable improvement in its English language, figure referencing, and clarity in explanations before publication.

Regarding the novelty of the research presented by the authors, I have to say that the research presents incremental but relevant novelty. While optimizing slurry composition is not new, the integration of experimental modeling with microstructural SEM analysis in the context of saturated sands provides new engineering insights and contributes to applied knowledge in slurry wall technology.

Therefore, I recommend minor revisions. With attention to these revisions—particularly the language, figure labeling, and clarity of analysis—the manuscript will be ready for publication.

Here are my comments:

  1. Widespread grammatical errors, awkward phrasing, and redundancy.
    e.g., Page 1, line 16: “stability.A multiple regression...” → should be “stability. A multiple regression...”
  2. Phrases like “play an effective wall protection effect” should be replaced with clearer alternatives such as “provide effective wall protection.”

  3. Replace inconsistent use of “slurry skin” with “slurry film” or “filter cake.”
  4. Ensure consistent and correct formatting of chemical notations: Na₂CO₃, CMC, etc.
  5. Long, unclear sentences in the Introduction and Results sections should be revised or split for clarity.
    e.g., Page 2, lines 34–42: The problem statement could be more concise and precise.
  6. Page 7, line 516: Spelling: “Box-Benhnken” is misspelled. It should be “Box–Behnken” (swap the ‘h’ and ‘n’). Please correct this spelling here and anywhere else it appears. Also, the phrasing “Box–Behnken orthogonal test method” can be simplified to “a Box–Behnken design (an RSM orthogonal design method)” to be more standard. Consider adding a brief description or citation for the Box–Behnken design for readers not familiar with it.
  7. Page 7, lines 518–526: This part defines factors A, B, C, D for the regression. Please ensure that the text clearly states what each factor is (it seems to: bentonite=A, clay=B, Na₂CO₃=C, CMC=D). Also, replace “the proportion, viscosity, sand content and pH” with “specific gravity, viscosity, sand content, and pH” for consistency (as noted before, avoid using “proportion” when you mean density or specific gravity). So line 523–526 should read: “At the same time, the specific gravity, viscosity, sand content, and pH — the main performance indices identified for the saturated sand layer slurry — were taken as the response values of the model.” This will align terminology with earlier sections.
  8. Figure 5 is used for both a flow chart and slurry film morphology images. Correct figure numbering and subfigure labels (a), (b), (c), etc.
  9. Ensure all figures and tables are referenced in order and properly described in the text.
  10. The film-forming test procedure needs more precise detail (e.g., dimensions, filtration criteria, pressure application).
  11. Table 6 and related text would benefit from a clearer explanation of group designations and purpose.
  12. In conclusion section, please distinguish between findings for optimal vs. suboptimal mix ratios.
  13. Also in conclusion section, revise phrases like “This law can provide theoretical guidance...” → use “These findings provide theoretical guidance…”
  14. Reference to standards/codes: You cited Chinese standards [26][27]. Make sure the references list includes them with proper English titles (if available) or at least transliteration. Also, reference [29] (used for test method) should be given in the list. Check that all references cited in the text (e.g., [1] and [29]) are present in the reference list and vice versa.

Author Response

Comment 1: 

Widespread grammatical errors, awkward phrasing, and redundancy. e.g., Page 1, line 16: “stability.A multiple regression...” → should be “stability. A multiple regression...”

Response 1:

Based on the grammar check performed by specialized software, we have further engaged a native English-speaking editor for comprehensive academic polishing of the manuscript. All revised sections have been highlighted in red throughout the text.

The concern pointed out by the reviewer has been revised in line 17 of the manuscript.

Comment 2: 

Phrases like “play an effective wall protection effect” should be replaced with clearer alternatives such as “provide effective wall protection.”

Response 2:

We agree with the reviewer's comment. The sentence has been polished and modified as follows (in lines 42-44):

Conventional bentonite-based slurries are particularly susceptible to the adverse effects of saturated sands, resulting in compromised wall protection performance [2].

Comment 3: 

Replace inconsistent use of “slurry skin” with “slurry film” or “filter cake.”

Response 3:

We appreciate the reviewer's careful review, and we have replaced all instances of “slurry skin” with “slurry film” throughout the manuscript (in line153).

Comment 4: 

Ensure consistent and correct formatting of chemical notations: Na2CO3, CMC, etc.

Response 4:

We have thoroughly checked and standardized the consistency and formatting of all chemical notations throughout the manuscript. Specifically:

  1. Unified all instances of “Soda ash”to “Na2CO3”;
  2. Added full names (e.g., Sodium Carboxymethyl Cellulose (CMC) and Polyacrylamide (PAM)) at their first occurrences (in line 74, 143);
  3. Included an "Abbreviations" section before the references to clarify all abbreviated terms (in line 726).

Comment 5: 

Long, unclear sentences in the Introduction and Results sections should be revised or split for clarity. e.g., Page 2, lines 34–42: The problem statement could be more concise and precise.

Response 5:

In response to the reviewers’ comments, we have thoroughly revised and improved lengthy or unclear sentences throughout the manuscript to enhance clarity and readability. The sentence at the beginning of the introduction proposed by the reviewer has been revised as follows (in lines 35-44):

Globally, bored pile technology is widely used for bridge foundation construction. However, in saturated sand layers with weak cementation, this method often leads to borehole instability issues such as wall shrinkage and collapse [1], significantly impacting construction schedules and long-term structural performance. Slurry wall protection serves as the primary technique for maintaining borehole stability during drilling operations. However, in saturated sand formations, multiple factors including soil characteristics, pore water pressure, borehole depth, and slurry film morphology collectively contribute to a complex mechanism. Conventional bentonite-based slurries are particularly susceptible to the adverse effects of saturated sands, resulting in compromised wall protection performance [2].

Comment 6: 

Page 7, line 516: Spelling: “Box-Benhnken” is misspelled. It should be “Box–Behnken” (swap the ‘h’ and ‘n’). Please correct this spelling here and anywhere else it appears. Also, the phrasing “Box–Behnken orthogonal test method” can be simplified to “a Box–Behnken design (an RSM orthogonal design method)” to be more standard. Consider adding a brief description or citation for the Box–Behnken design for readers not familiar with it.

Response 6:

Thank you for your meticulous review. We have carefully addressed your comments as follows:

  1. Spelling Correction:

All instances of “Box-Benhnken” have been corrected to “Box-Behnken” throughout the manuscript.

  1. Terminology Standardization:

The phrasing has been updated to “a Box-Behnken design (a response surface methodology orthogonal design)” as suggested (in lines 273-274).

  1. Methodological Clarification:

A brief description of the Box-Behnken design's advantages has been added to lines 259-262, supported by citations to established literature (References [32]).

The Box-Behnken design is an efficient response surface methodology that requires fewer experimental runs than full factorial designs, while maintaining the ability to estimate quadratic effects [32]. This design is particularly suitable for optimizing multiple slurry parameters.

[32]Box, G.E.P.; Behnken, D.W. Some new three level designs for the study of quantitative variables. Technometrics 1960, 2, 455-475.

Comment 7: 

Page 7, lines 518–526: This part defines factors A, B, C, D for the regression. Please ensure that the text clearly states what each factor is (it seems to: bentonite=A, clay=B, Na₂CO₃=C, CMC=D). Also, replace “the proportion, viscosity, sand content and pH” with “specific gravity, viscosity, sand content, and pH” for consistency (as noted before, avoid using “proportion” when you mean density or specific gravity). So line 523–526 should read: “At the same time, the specific gravity, viscosity, sand content, and pH — the main performance indices identified for the saturated sand layer slurry — were taken as the response values of the model.” This will align terminology with earlier sections.

Response 7:

Thank you for highlighting these important clarity issues. We have implemented the following improvements:

  1. Explicit factor definitions:

Added a clear listing of variables A–D with their corresponding materials and units (Bentonite=A, Clay=B, etc.) in Section 2.4. The modified content is as follows (in lines 275-277):

The independent variables were defined as: A: Bentonite content (g), B: Clay content (g), C: Na₂CO₃ content (g), D: Carboxymethyl cellulose (CMC) content (g).

  1. Terminology consistency:

Replaced “proportion” with “specific gravity” throughout the text to align with the standard metrics in slurry studies. Following the reviewer's suggestion, we have updated the content in lines 277-279 as follows:

At the same time, the specific gravity, viscosity, sand content, and pH — the main performance indices identified for the saturated sand layer slurry — were taken as the response values of the model.

Comment 8: 

Figure 5 is used for both a flow chart and slurry film morphology images. Correct figure numbering and subfigure labels (a), (b), (c), etc.

Response 8:

We sincerely appreciate your careful review and valuable feedback. We have thoroughly checked all figures in the manuscript and made the following corrections:

  1. Figure Numbering:

The duplicate “Figure 5” has been renumbered. The slurry film morphology images (originally labeled as two separate “Figure 5”) are now correctly labeled as Figure 6.

  1. Subfigure Labels:

The repeated “(a)” in the caption of Figure 6 (previously mislabeled as Figure 5) has been corrected to “(a), (b), (c)” to match the three subfigures.

  1. Cross-verification:

We have double-checked all figure citations and ensured alignment between labels, captions, and in-text references.

Comment 9: 

Ensure all figures and tables are referenced in order and properly described in the text.

Response 9:

We have checked and confirmed the tables, figure numbers, and positions in the manuscript.

Comment 10: 

The film-forming test procedure needs more precise detail (e.g., dimensions, filtration criteria, pressure application).

Response 10:

Thank you for the careful review by the reviewer. We have provided a more detailed description of the experimental method in section 3.1.2 (Lines 457-486). The revised content is as follows:

The film-forming test procedure was conducted using a rigorously controlled experimental protocol [36]. The slurry was prepared according to pre-set standards (as specified in Table 6) and allowed to hydrate for 24 h to ensure complete particle dispersion. The test apparatus consisted of a transparent cylindrical cell with precise dimensions (600 mm height × 391 mm internal diameter × 20 mm wall thickness), featuring a multi-layer construction. The base layer comprised graded gravel (100 mm thickness, 2–5 mm particle size) for filtration control, overlain by a 200 mm thick sand/silt-clay test stratum that was compacted in 30–40 mm lifts to achieve a dry density of 1.65 g/cm3, accurately simulating field conditions. Saturation was achieved through upward water injection at a controlled rate of 0.05 L/min until full submersion, followed by a 12-h stabilization period to ensure ≥ 95% saturation, confirmed by the absence of air bubble emissions for a minimum 2-h observation window.

For the actual testing phase, pre-hydrated slurry (50 mm thickness) was carefully introduced through a 10-mm diameter injection port at a flow rate of 5 mL/s to prevent stratum disturbance. The system was then sealed using flanged connections with rubber gaskets and subjected to a 0.05 MPa preliminary pressure test for 10 min to verify airtight integrity. The formal testing protocol implemented four precisely controlled pressure stages (0.1, 0.2, 0.3, and 0.4 MPa), each featuring a 20-second linear ramp-up phase followed by a 160-second steady-state maintenance period where pressure fluctuations were maintained within ± 1% of target values. Film formation was determined to occur when two concurrent criteria were met: (1) the filtrate flow rate slope measured by precision electronic balance (±0.01 g accuracy) decreased to ≤0.75 mL/s, and (2) visual stabilization of the slurry-sand interface was confirmed through high-definition video monitoring. Upon meeting these criteria, the system was carefully depressurized (5-second valve opening sequence) and the resulting film thickness was measured at five radial positions using laser displacement metrology (±0.1 mm resolution). Representative samples (5×5 mm) were extracted from both central and peripheral regions of the formed film for subsequent SEM analysis, which included gold sputter coating and imaging at 500–10,000× magnification to characterize microstructural features.

Comment 11: 

Table 6 and related text would benefit from a clearer explanation of group designations and purpose.

Response 11:

Thank you for your valuable suggestion. We have revised the manuscript to clarify the experimental design and purpose of the grouped tests in Table 6. The modifications include:

  1. Enhanced Table Title & Footnotes:

Original title: Table 6. Slurry optimization ratio parameters.

Revised to: Table 6. Experimental groups for slurry optimization: additive combinations and performance parameters

Added footnote:

Groups a–f represent systematic combinations of clay and CMC additives to evaluate their synergistic effects on slurry viscosity and specific gravity. Baseline Group (a) contains no additives, while Groups b–f incrementally introduce clay (0–220 g) and CMC (0–2 g) to isolate their individual and combined impacts.

  1. Expanded Methodology Section (Section 3.1.1):

Added a new paragraph before Table 6 (in lines 444-449):

The six experimental groups (a–f) were designed to decouple the effects of clay (particle filling) and CMC (viscosity modification) on slurry performance. Group a served as the additive-free control, while Groups b–f progressively introduced: Clay (0–220 g) to enhance particle gradation and pore-filling capacity, CMC (0–2 g) to improve colloidal stability and fluid viscosity. 

Comment 12: 

In conclusion section, please distinguish between findings for optimal vs. suboptimal mix ratios.

Response 12:

We have added relevant content in the conclusion section, as follows:

In lines 678-679: Suboptimal ratios (e.g., clay:CMC > 220:1) showed diminished returns, with viscosity plateauing beyond 2g CMC due to particle agglomeration.

In lines 687-689: In contrast, high-clay formulations (e.g., 220:0, Group e) increased specific gravity but reduced film uniformity, while CMC-deficient groups (e.g., 110:0, Group b) exhibited 20‒25% higher filtration loss.

In lines 703-705: Suboptimal mixes (e.g., excessive clay without CMC) required 10‒15% more material to achieve comparable stability, negating cost benefits.

Comment 13: 

Also in conclusion section, revise phrases like “This law can provide theoretical guidance...” → use “These findings provide theoretical guidance…

Response 13:

We have rephrased the conclusion section (in lines 671-715), and the revised part related to this comment is as follows:

These results provide both theoretical understanding and practical guidelines for slurry design in challenging hydrogeological conditions.

Comment 14: 

Reference to standards/codes: You cited Chinese standards [26][27]. Make sure the references list includes them with proper English titles (if available) or at least transliteration. Also, reference [29] (used for test method) should be given in the list. Check that all references cited in the text (e.g., [1] and [29]) are present in the reference list and vice versa.

Response 14:

We sincerely appreciate the reviewer’s careful reading and constructive suggestions. The following revisions have been made to address this comment:

  1. Standard [28] and [29]:

Added official English translations of the Chinese standard titles in the References section:

[28]Ministry of Geology and Mineral Resources of China. Construction Specifications for Drilled Cast-in-Place Piles (DZ/T 0155-95). Beijing, 1995 (in Chinese).

[29]Ministry of Housing and Urban-Rural Development of China. Technical Specification for Large-Diameter Belled Cast-in-Place Pile foudation (JGJ/T 225-2010). Beijing, 2011 (in Chinese).

  1. Reference [36]:

The citation for [36] has now been added to the References section with full details:

[36]Ye, W.T.; Wang, J.Y.; Fu, L.L.; Zhou, S.H.; Ning, J.W. Laboratory test and characteristic of filter film formation of slurry shield in medium-coarse sand stratum in Fuzhou. Chinese Journal of Rock Mechanics and Engineering, 2018, 37(5): 1260–1269 (in Chinese with English abstract).

  1. Cross-check:

Verified that all in-text citations are now matched with entries in the References list, and no uncited references remain.

Round 2

Reviewer 1 Report

Comments and Suggestions for Authors

The authors have effectively addressed all my previous comments. The revised manuscript is meaningfully improved, and I recommend it for publication in its current form.

Author Response

Commet 1:

The authors have effectively addressed all my previous comments. The revised manuscript is meaningfully improved, and I recommend it for publication in its current form.

Response 1:

We are very grateful to the reviewer for this positive feedback and for their time and effort throughout the review process. We are delighted that the revisions have addressed their concerns satisfactorily. We thank the reviewer once again for their constructive comments, which have significantly improved the quality of our manuscript.

Reviewer 2 Report

Comments and Suggestions for Authors

Thank you very much for your extremely meticulous revisions. I truly respect your attitude and dedication to conveying your findings clearly and reliably to readers. You have addressed all comments perfectly, and I have confirmed that the responses are well-reflected in the revised manuscript. I am confident that your revisions have made the paper significantly more readable and valuable.

Below are a few minor suggestions for your consideration:

(1) Line 199
Before: Bentonite: Sourced from Zhejiang Hongyu...
Suggested revision: Bentonite (sodium type): Sourced from Zhejiang Hongyu...

(2) Line 289
Before: pH values are whole numbers as per colorimetric strip limitations.
Suggested revision: pH values are whole numbers due to detection limitations (±0.5) of the colorimetric strip method.

(3) Line 317
Please italicize the parameters p and R².

(4) Table 7
To maintain consistency with the main text where "mL" is used, please change all instances of "ml" in the table to "mL".

Thank you again.

Author Response

Comment 1:

Line 199

Before: Bentonite: Sourced from Zhejiang Hongyu...

Suggested revision: Bentonite (sodium type): Sourced from Zhejiang Hongyu...

Response 1:

Thank you for your careful review. We have made the revision as suggested at Line 198, with the modified text highlighted in blue.

Comment 2:

Line 289

Before: pH values are whole numbers as per colorimetric strip limitations.

Suggested revision: pH values are whole numbers due to detection limitations (±0.5) of the colorimetric strip method.

Response 2:

We appreciate the reviewer’s suggestion and have revised lines 288-289 accordingly.

Comment 3:

Line 317

Please italicize the parameters p and R2.

Response 3:

Thank you for the reviewer’s comment. We have now italicized ‘p’ and ‘R2’ in line 317 and have checked the entire manuscript for similar issues.

Comment 4:

Table 7

To maintain consistency with the main text where "mL" is used, please change all instances of "ml" in the table to "mL".

Response 4:

We appreciate the reviewer's attention to detail. We have updated the units in Table 7 from "ml" to "mL" to match the consistent usage in the main text. Additionally, we have performed a quick check to ensure that "mL" is used consistently. 

Reviewer 3 Report

Comments and Suggestions for Authors

The authors have carefully revised the manuscript in line with the reviewers’ suggestions. The scientific content is now well-presented and provides a valuable contribution to the field. I recommend the manuscript for acceptance.

Author Response

Comment 1:

The authors have carefully revised the manuscript in line with the reviewers’ suggestions. The scientific content is now well-presented and provides a valuable contribution to the field. I recommend the manuscript for acceptance.

Response 1:

We thank the reviewer for their positive assessment and for recommending our manuscript for acceptance. We are delighted that the revisions have addressed their concerns and that they find the scientific contribution valuable. Their insightful comments have greatly improved the quality of our work.